# Elementary sensory-motor transformations underlying olfactory navigation in walking fruit-flies

Efrén Álvarez-Salvado[1], Angela M Licata[1], Erin G Connor[2], Margaret K McHugh[2], Benjamin MN King[1], Nicholas Stavropoulos[1], Jonathan D Victor[3,4], John P Crimaldi[2], Katherine I Nagel[1]*

[1]Neuroscience Institute, New York University Langone Medical Center, New York, United States; [2]Department of Civil, Environmental and Architectural Engineering, University of Colorado Boulder, Boulder, United States; [3]Institute for Computational Biomedicine, Weill Cornell Medical College, New York, United States; [4]Feil Family Brain and Mind Research Institute, Weill Cornell Medical College, New York, United States

**Abstract** Odor attraction in walking *Drosophila melanogaster* is commonly used to relate neural function to behavior, but the algorithms underlying attraction are unclear. Here, we develop a high-throughput assay to measure olfactory behavior in response to well-controlled sensory stimuli. We show that odor evokes two behaviors: an upwind run during odor (ON response), and a local search at odor offset (OFF response). Wind orientation requires antennal mechanoreceptors, but search is driven solely by odor. Using dynamic odor stimuli, we measure the dependence of these two behaviors on odor intensity and history. Based on these data, we develop a navigation model that recapitulates the behavior of flies in our apparatus, and generates realistic trajectories when run in a turbulent boundary layer plume. The ability to parse olfactory navigation into quantifiable elementary sensori-motor transformations provides a foundation for dissecting neural circuits that govern olfactory behavior.
DOI: https://doi.org/10.7554/eLife.37815.001

*For correspondence:
katherine.nagel@nyumc.org

**Competing interests:** The authors declare that no competing interests exist.

## Introduction

Fruit-flies, like many animals, are adept at using olfactory cues to navigate toward a source of food. Because of the genetic tools available in this organism, *Drosophila melanogaster* has emerged as a leading model for understanding how neural circuits generate behavior. Olfactory behaviors in walking flies lie at the heart of many studies of sensory processing (*Root et al., 2008*; *Su et al., 2012*), learning and memory (*Aso et al., 2014*; *Owald et al., 2015*), and the neural basis of hunger (*Root et al., 2011*; *Tsao et al., 2018*). However, the precise algorithms by which walking flies locate an odor source are not clear.

Algorithms for olfactory navigation have been studied in a number of species, and can be broadly divided into two classes, depending on whether the organisms typically search in a laminar environment or in a turbulent environment. In laminar environments, odor concentration provides a smooth directional cue that can be used to locate the odor source. Laminar navigators include bacteria (*Brown and Berg, 1974*), nematodes (*Pierce-Shimomura et al., 1999*), and *Drosophila* larvae (*Gomez-Marin et al., 2011*; *Gershow et al., 2012*). In each of these organisms, a key computation is detection of temporal changes in odor concentration, which drives changes in the probability of re-orientation behaviors. In turbulent environments, odors are transported by the instantaneous structure of air or water currents, forming plumes with complex spatial and temporal structure

**eLife digest** All kinds of animals use their sense of smell to find things. Doing this is difficult because odors in air travel as plumes, which meander downwind and break apart. Scientists are interested in learning the rules that animals use to decipher these odor signals and trace them back to their source. For example, do animals use patterns of timing in the odor, differences between smell at the two nostrils, or the direction of the wind? Scientists would also like to know how animal's brain circuits decipher this information.

Tiny fruit flies make a good model for studying the way animals detect odors because scientists have already learned a great deal about how their brains work. There are also many tools available to help scientists study the brain circuits of fruit flies.

Now, Álvarez-Salvado et al. show that fruit flies use multiple senses to track odors to their source. In the experiments, fruit flies that were blind and could not fly were placed in tiny wind tunnels and their behavior in response to a smell or no smell in the tunnel was carefully documented. When the flies detected an odor, they turned to face the wind using their antennae to detect wind direction and run toward it. When flies lost track of an odor they began to search for it at the spot where they last smelled it. Next, Álvarez-Salvado et al. created a computer model that recreated the flies' behavior and was able to find the odor source as well as real flies. The model added together these basic behaviors to successfully recreate the flies' odor-search strategy.

Other animals are often better than humans at finding odor sources. As a result, people use pigs to find truffles and dogs to find lost hikers. The computer model Álvarez-Salvado et al. developed might help design robots that can search for truffles, hikers, or landmines, without risking the lives of animals. It might also be useful for designing autonomous vehicles that must respond to many types of information in changing environments to make decisions.

DOI: https://doi.org/10.7554/eLife.37815.002

(*Crimaldi and Koseff, 2001*; *Crimaldi et al., 2002*; *Webster and Weissburg, 2001*). Within a turbulent plume, odor fluctuates continuously, meaning that instantaneous concentration gradients do not provide simple information about the direction of the source . Navigation in turbulent environments has been studied most extensively in moths (*Kennedy and Marsh, 1974*; *David et al., 1983*; *Baker, 1990*; *Kuenen and Carde, 1994*; *Rutkowski et al., 2009*) but has also been investigated in flying adult *Drosophila* (*van Breugel and Dickinson, 2014*) and marine plankton (*Page et al., 2011*). In these organisms, the onset or presence of odor drives upwind or upstream orientation, while loss of odor drives casting orthogonal to the direction of flow. An important distinction between laminar and turbulent navigation algorithms is that the former depend only on the dynamics of odor concentration, while the latter rely also on measurements of flow direction derived from mechanosensation or optic flow (*Cardé and Willis, 2008*). Also unclear is the role of temporal cues in turbulent navigation. Several studies have suggested that precise timing information about plume fluctuations might be important for navigation (*Baker, 1990*; *Mafra-Neto and Cardé, 1994*), or that algorithms keeping track of the detailed history of odor encounters may promote chemotaxis (*Vergassola et al., 2007*), but the relationship between odor dynamics and olfactory behaviors has been challenging to measure experimentally (*Pang et al., 2018*).

In comparison to these studies, olfactory navigation in walking flies has not been studied as quantitatively. A walking fly in nature will encounter an odor plume that is developing close to a solid boundary. Such plumes are broader, exhibit slower fluctuations, and allow odor to persist further downwind from the source, compared to the airborne plumes encountered by flying organisms (*Crimaldi and Koseff, 2001*; *Crimaldi et al., 2002*; *Webster and Weissburg, 2001*). Navigational strategies in these two environments might therefore be different. In laboratory studies, walking flies have been shown to turn upwind when encountering an attractive odor (*Flügge, 1934*; *Steck et al., 2012*), and downwind when odor is lost (*Bell and Wilson, 2016*). However, flies can also stay within an odorized region when wind cues provide no direction information, by modulating multiple parameters of their locomotion (*Jung et al., 2015*). Finally, walking flies have been shown to turn towards the antenna that receives a higher odor concentration (*Borst and Heisenberg, 1982*; *Gaudry et al.,*

*2013*). It is not clear how these diverse motor programs work together to promote navigation toward an attractive odor source in complex natural environments.

Here, we set out to define elementary sensory-motor transformations that underlie olfactory navigation in walking fruit flies. To this end, we designed a miniature wind-tunnel paradigm that allows us to precisely control the wind and odor stimuli delivered to freely walking flies. Using this paradigm, we show that flies, like other organisms, navigate through distinct behavioral responses to the presence and loss of odor. During odor, flies increase their ground speed and orient upwind. Following odor loss, they reduce their ground speed and increase their rate of turning. By blocking antennal wind sensation, we show that mechanosensation is required for the directional components of these behaviors, while olfaction is sufficient to induce changes in ground speed and turning. This implies that olfactory navigation is driven by both multi-modal and unimodal sensori-motor transformations. We next used an array of well-controlled dynamic stimuli to define the temporal features of odor stimuli that drive upwind orientation and turn probability. We find that behavioral responses to odor are significantly slower than peripheral sensory encoding, and are driven by an integration of odor information over several hundred milliseconds (for upwind orientation) and several seconds (for turn probability).

To understand how these elementary responses might promote navigation in a complex environment, we developed a simple computational model of how odor dynamics and wind direction influence changes in forward and angular velocity. We show that this model can recapitulate the mean behavior of flies responding to a pulse stimulus, as well as the variability in response types observed across flies. Finally, we examine the behavior of our model in a turbulent odor plume measured experimentally in air, finding that its performance is comparable to that of real flies in the same environment. These simulations suggest that integration over time may be a useful computational strategy for navigating in a boundary layer plume, allowing flies to head upwind more continuously in the face of odor fluctuations, and to generate re-orientations clustered at the plume edges. Moreover, they suggest that multiple independent forms of sensing —flow sensing, temporal sensing, and spatial sensing— can work cooperatively to promote attraction to an odor source. Our description of olfactory navigation algorithms in walking flies, and the resulting computational model, provide a quantitative framework for analyzing how specific sensory-motor transformations contribute to odor attraction in a complex environment, and will facilitate the dissection of neural circuits contributing to olfactory behavior.

## Results

### ON and OFF responses to odor in a miniature wind-tunnel paradigm

To investigate the specific responses underlying olfactory navigation, we developed a miniature wind-tunnel apparatus in which we could present well-controlled wind and odor stimuli to walking flies (*Figure 1A and B* and Materials and methods). Flies were placed in rectangular arenas, where they were exposed to a constant flow of filtered, humidified air, defining the wind direction. Into this airflow we injected pulses of odor with rapid onset and offset kinetics, producing a front of odor that was transported down the arena at 11.9 cm/s. The time courses of odor concentration and air speed inside the behavioral arena were measured using a photo-ionization detector (PID) and an anemometer (*Figure 1E* ). Because flies were free to move about the chamber, and because the odor from takes about 1 s to advect down the arena, flies encountered and lost the odor at slightly different times. We therefore used PID measurements made a several locations in the arena to warp our behavior data to the exact times of odor onset and offset (see Materials and methods, *Figure 1—figure supplement 1*). We used genetically blind flies (*norpA*[36] mutants) in order to remove any possible contribution of visual responses. Flies were starved 5 hr prior to the experiment, and were tested for approximately 2 hr (from ZT 2–4), in a series of 70 second-long trials with blank (wind only) and odor trials randomly interleaved.

We observed that in the presence of 10% apple cider vinegar (ACV), flies oriented upwind, and moved faster and straighter (*Figure 1C*, magenta traces). This 'ON' response peaked 4.4±2.5 s after odor onset, but remained as long as odor was present. Following odor offset, flies exhibited more tortuous and localized trajectories (*Figure 1C*, cyan traces). This 'OFF' response resembles local search behavior observed in other insects (*Willis et al., 2008*), and persisted for tens of seconds

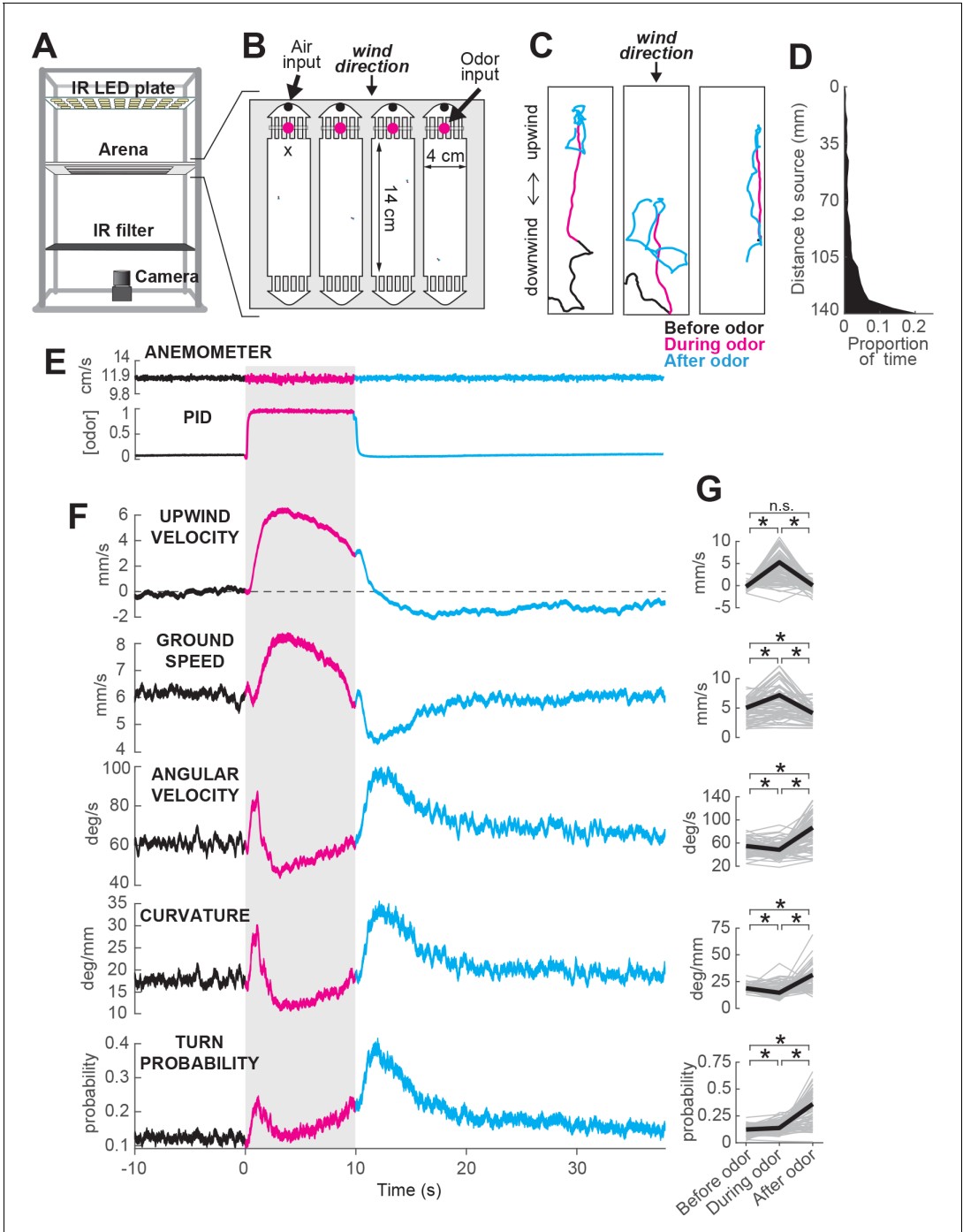

**Figure 1.** ON and OFF responses to an attractive odor pulse. (A) Schematic of the behavioral apparatus (side view) showing illumination and imaging camera. (B) Schematic of the behavioral arena (top view) showing four behavior chambers and spaces to direct air and odor through the apparatus. Dots mark air and odor inputs. Black cross: site of wind and odor measurements in E. (C) Example trajectories of three different flies before (black), during (magenta) and after (cyan) a 10 s odor pulse showing upwind runs during odor and search after odor offset. (D) Distribution of fly positions on trials with wind and no odor; flies prefer the downwind end of the arena. (E) Average time courses of wind (top; anemometer measurement; n = 10) and odor (bottom; PID measurement normalized to maximal concentration; n = 10) during 10 s odor trials. Measurements were made using 10% ethanol at the arena position shown in B. (F) Calculated parameters of fly movement averaged across flies (mean±SEM; n = 75 flies, 1306 trials; see Materials and methods). Traces are color coded as in C. Gray-shaded area: odor stimulation period (ACV 10%). All traces warped to estimated time of odor encounter and loss prior to averaging. Small deflections in ground speed near the time of odor onset and offset represent a brief stop response to the click of the odor valves (see *Figure 3—figure supplement 1*). (G) Average values of motor parameters in F for each fly for periods before (−30 to 0 s), during (2 to 3 s) and after (11 to 13 s) the odor. Gray lines: data from individual flies. Black lines: group average. Horizontal lines with asterisk:

*Figure 1 continued on next page*

*Figure 1 continued*

Statistically significant changes in a Wilcoxon signed rank paired test after correction for multiple comparisons using the Bonferroni method (see Materials and methods for p values). n.s.: not significant.

DOI: https://doi.org/10.7554/eLife.37815.003

The following figure supplements are available for figure 1:

**Figure supplement 1.** Warping method corrects for differences in odor encounter timing as a function of position within the arena.

DOI: https://doi.org/10.7554/eLife.37815.004

**Figure supplement 2.** Variability between individuals in responses to an odor pulse.

DOI: https://doi.org/10.7554/eLife.37815.005

**Figure supplement 3.** Sighted flies show ON and OFF responses to odor.

DOI: https://doi.org/10.7554/eLife.37815.006

after odor offset. These two responses are usually readily perceptible and distinguishable by observing the movements of flies during an odor pulse (*Figure 1C*, *Video 1*). On trials without odor, flies tended to aggregate at the downwind end of the arena (*Figure 1D*).

To analyze these responses quantitatively, we first noted that flies alternated between periods of movement and periods of immobility (*Figure 3—figure supplement 1A–B*). To focus on the active responses of flies, we considered in our analyses only those periods in which flies were moving, and we established a threshold of 1 mm/s below which flies were considered to be stationary (see Materials and methods). Then, we analyzed how flies' movements changed in response to an odor pulse by extracting a series of motor parameters (*Figure 1F*, see Materials and methods). We computed each measure both as a function of time (*Figure 1F*) and on a fly-by-fly basis for specific time intervals before, during, and after the odor presentation (*Figure 1G*).

During odor presentation, upwind velocity (i.e. speed of flies along the longitudinal axis of the arenas) and ground speed both increased significantly, while angular velocity and curvature (i.e. ratio between angular velocity and ground speed) decreased after an initial peak. This resulted in the straighter trajectories observed during odor; the initial peak observed in angular velocity and curvature corresponds to big turns performed by flies to orient upwind after odor onset. Following odor offset, angular velocity increased, while ground speed decreased, resulting in the increased curvature characteristic of local search (*Figure 1F,G*). Since an increase in probability of reorientation has been traditionally identified as a hallmark of localized search (*Brown and Berg, 1974*; *Pierce-Shimomura et al., 1999*; *Gomez-Marin et al., 2011*; *Gershow et al., 2012*), we calculated the turn probability of flies in our arena as a binarized version of curvature around a threshold of 20 deg/mm. Indeed, turn probability increased as well after odor offset (*Figure 1F,G*). Upwind velocity also became negative after odor offset, although this response was weaker than the upwind orientation during odor, and peaked later than the changes in ground speed and curvature.

Although most of the flies we tested showed ON and OFF responses as described above, we observed considerable variability between individuals (*Figure 1—figure supplement 2*). Individuals varied in the strength of their odor responses, with some flies exhibiting strong upwind orientation and search, while others showed little odor-evoked modulation of behavior (*Figure 1—figure supplement 2A–C*). Motor parameters from the same individual in different trials were correlated, whereas parameters randomly selected from different individuals were

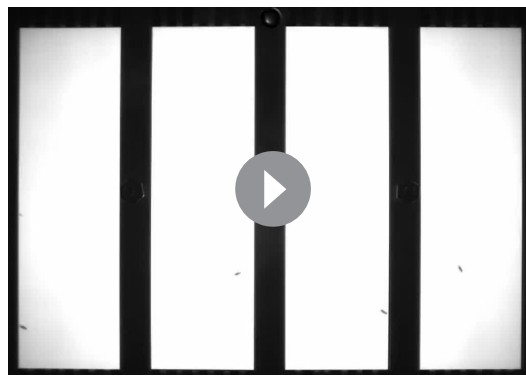

**Video 1.** Behavior of four flies in response to an ACV 10% pulse. The time of the odor stimulus is signaled by the green dot appearing at the top of the image. Flies start to move upwind shortly after the start of the stimulus (partly due to the time it takes for the odor front to reach their respective positions), and they stop advancing upwind after the odor is gone and engage in a more localized search behavior. Air and odor move from the top of the image towards the bottom at 11.9 cm/s.

DOI: https://doi.org/10.7554/eLife.37815.007

not (*Figure 1—figure supplement 2D*). Thus, the movement parameters of the 'average fly' depicted in *Figure 1* underestimate the range of search behaviors shown by individuals, with particular flies exhibiting both much stronger and much weaker ON and OFF responses. There was a slight tendency for responses to be weaker during the first few trials; afterwards, this behavior was stable (on average) across the entire experimental session (*Figure 1—figure supplement 2F*). Sighted flies of the same genetic background also showed ON and OFF responses (*Figure 1—figure supplement 3*), with increases in upwind velocity and ground speed during odor, and increases in angular velocity and decreased ground speed after odor offset. However, the increase in angular velocity appeared to be weaker, on average, in these flies.

Together, these data indicate that apple cider vinegar drives two distinct behavioral responses: an ON response consisting of upwind orientation coupled with faster and straighter trajectories, and an OFF response consisting of slower and more curved trajectories.

## Local search is driven purely by odor dynamics

We next asked whether any change in behavior could be produced by odor in the absence of wind information. Previous studies have found that optogenetic activation of $orco^+$ neurons did not elicit attraction (*Suh et al., 2007*), unless wind was present (*Bell and Wilson, 2016*). However, modulation of gait parameters by odor has also been observed when the wind is directed perpendicular to the plane of the arena (*Jung et al., 2015*). To ask whether walking flies could respond to odor in the absence of wind, we stabilized the third segment of the antennae using a small drop of UV glue. Fruit flies sense wind direction using stretch receptors that detect rotations of the third antennal segment (*Yorozu et al., 2009*). This manipulation therefore renders flies 'wind-blind' (*Budick et al., 2007*; *Bhandawat et al., 2010*).

We found that wind-blind flies showed severely impaired directional responses to odor and wind. Upwind velocity was not significantly modulated either during the odor or after (*Figure 2A–B*, top). Indeed, odor-induced runs in different directions (either up- or downwind or sideways) could be observed in individual trajectories (*Figure 2C*). In addition, the downwind positional bias seen in the absence of odor was reduced (*Figure 2D*). The average arena position of wind-blind flies on no-odor trials was no different from that of intact flies in the absence of wind (*Figure 2D*). Thus, antennal wind sensors are critical for the oriented components of olfactory search behavior.

However, wind-blind flies still responded to odor by modulating their ground speed and angular velocity. Wind-blind flies increased their curvature after odor offset and also increased their ground speed during odor (*Figure 2B*). These changes can be seen in the examples shown in *Figure 2C*, where flies adopt somewhat straighter trajectories during odor, and exhibit local search behavior following odor offset. These results imply that odor can directly modulate gait parameters to influence navigation in the absence of wind. Together these experiments show that olfactory navigation depends both on multimodal processing (odor-gated upwind orientation), and on direct transformation of odor signals into changes in ground speed and curvature.

## ON and OFF responses to dynamic stimuli

Because natural odor stimuli are highly dynamic, we next asked what features of the odor signal drive ON and OFF responses. To address this question, we presented flies with a variety of dynamically modulated stimuli. We focused our analysis on upwind velocity and turn probability, as measures of the ON and OFF response, respectively, as these parameters provided the highest signal-to-noise ratio.

We first looked at how ON and OFF behaviors depended on the concentration of the odor stimulus. In these experiments, different groups of flies were exposed to square pulses of apple cider vinegar at dilutions of 0.01%, 0.1%, 1% and 10% (*Figure 3A–B*). We found that both upwind velocity during odor and turn probability after offset grew with increasing odor concentration between 0.01% and 1%, but saturated or even decreased at 10% (*Figure 3A–B*). These responses were well fit by a Hill function with a dissociation constant $\kappa_d$ of 0.072% (for ON) and and 0.127% (for OFF; *Figure 3A and B*, left and right insets). The fitted Hill coefficient was very close to 1 (1.03 for ON and 1.06 for OFF). A saturating Hill function nonlinearity is to be expected from odor transduction kinetics, and has been found to describe encoding of odor stimuli by peripheral olfactory receptor neurons (*Kaissling et al., 1987*; *Nagel and Wilson, 2011*; *Gorur-Shandilya et al., 2017*;

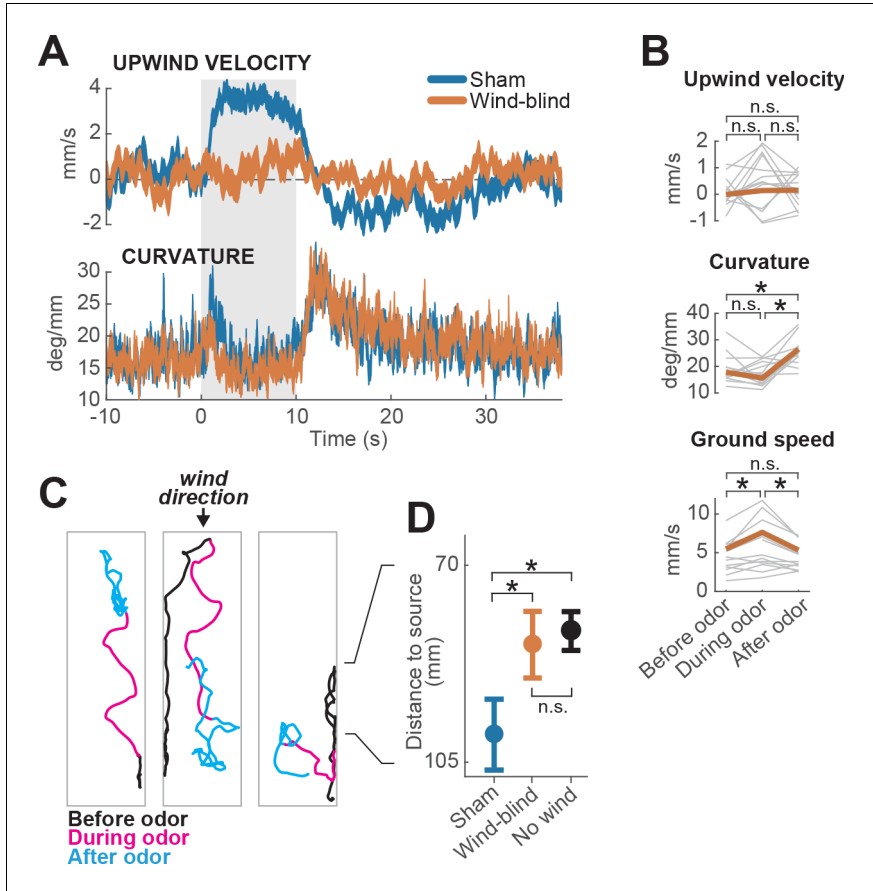

**Figure 2.** Multimodal and unimodal contributions to olfactory behavior. (A) Stabilization of the antennae abolishes odor-evoked changes in upwind velocity but not curvature. Traces show mean±SEM for wind-blind (n = 13 flies, 240 trials) and sham-treated flies (n = 15 flies, 217 trials; see Materials and methods) (B) Mean values of upwind velocity, curvature and ground speed in wind-blind flies during periods before, during, and after the odor pulse (time windows as in *Figure 1G*). Gray lines: data from individual wind-blind flies. Orange lines: group average. Horizontal lines with asterisk: statistically significant changes in a Wilcoxon signed rank paired test after correction for multiple comparisons using the Bonferroni method (see Materials and methods for p values). n.s.: not significant. (C) Example trajectories of three different wind-blind flies before (black), during (magenta) and after (cyan) the odor pulse. Note different orientations relative to wind during the odor. (D) Antenna stabilization decreases preference for the downwind end of the arena on trials with wind and no odor. Blue: average (±SEM) arena position of sham-treated flies on trials with wind and no odor. Orange: average position of wind-blind flies in the same stimulus condition. Black: Average position of intact (not-treated) flies in the absence of both odor and wind (n = 23 flies, 1004 trials). The average arena position of wind-blind flies did not differ significantly from that of no-wind flies (p=0.93). Sham-treated flies spent significantly more time downwind than wind-blind (p=0.04) or intact flies in the absence of wind (p=0.0027). Horizontal lines with asterisk: statistically significant changes in a Wilcoxon rank sum test (*alpha* = 0.05). n.s.: non-significant. Black lines between C and D provided for reference of dimensions in D.

DOI: https://doi.org/10.7554/eLife.37815.008

*Schulze et al., 2015*), and central olfactory projection neurons (*Olsen et al., 2010*). A decrease in response at the highest intensities could arise from inhibitory glomeruli that are recruited at higher odor intensity, as has been described in *Semmelhack and Wang, 2009*).

We next wondered whether OFF behaviors could be elicited by gradual decreases in odor concentration, as turning behavior in gradient navigators is sensitive to the slope of odor concentration (*Brown and Berg, 1974*; *Pierce-Shimomura et al., 1999*). To perform this experiment, we used proportional valves to deliver a pulse of saturating concentration (10% ACV), that then decreased linearly over a period of 2.5, 5 or 10 s (*Figure 3C–D*, Materials and methods). We observed that turn

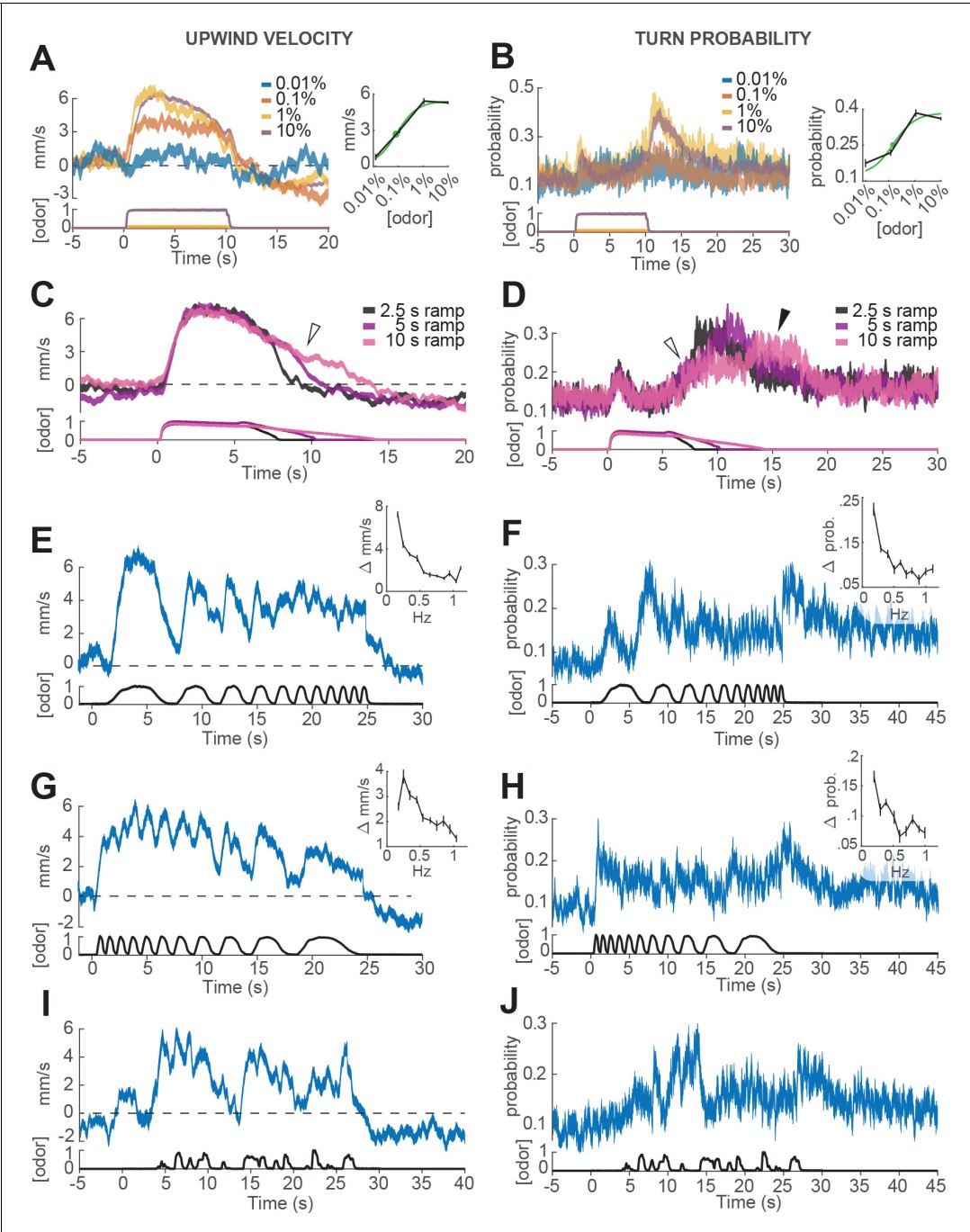

**Figure 3.** Responses of walking flies to dynamic odor stimuli. (A) Upwind velocity (left, top traces; average±SEM) of different groups of flies responding to a 10 s pulse of ACV at dilutions of 0.01% (n = 13 flies, 147 trials), 0.1% (n = 19 flies, 304 trials), 1% (n = 18 flies, 302 trials) and 10% (n = 75 flies, 1306 trials). Left-bottom traces show PID measurements using ethanol (max concentration 10%), normalized to maximal amplitude. Right inset: mean upwind velocity during odor (2 to 3 s) as a function of odor concentration (black; mean±SEM), and fitted Hill function (green; green dot: $\kappa_d$=0.072%). (B) Turn probability calculated from the same data. Right inset black traces: mean turn probability after odor (11 to 13 s). $\kappa_d$=0.127% for fitted Hill function (green). (C) Upwind velocity (average±SEM) in response to stimuli with off-ramps of 2.5 (n = 38 flies, 528 trials), 5 (n = 38 flies, 567 trials) and 10 (n = 35 flies, 557 trials) seconds duration. Bottom traces: PID signals of the same stimuli using ethanol. (D) Same as C, showing turn probability from the same data sets. White arrows in C and D show elevated upwind velocity and turn probability that co-occur during a slow off-ramp. Black arrow in D: peak turn probability response at the foot of the off-ramp. (E) Upwind velocity (mean±SEM; n = 31 flies, 346 trials) in response to an ascending frequency sweep stimulus. Bottom trace: PID signal of the stimulus, measured using ethanol. Right inset: average (±SEM) modulation of upwind velocity as a function of frequency in each stimulus cycle (see Materials and methods). (F) Same as E for turn probability calculated from the same data. Right inset: modulation of turn probability as a function of frequency. (G) Equivalent to E, showing responses to a descending frequency

*Figure 3 continued on next page*

*Figure 3 continued*

sweep (n = 33 flies, 345 trials). In the inset, the first high-frequency cycle was left out of the analysis. (H) Same as G for turn probability calculated from the same data. (I) Equivalent to G, showing responses to a simulated 'plume walk' (n = 30 flies, 393 trials). (J) Same as I for turn probability calculated from the same data.

DOI: https://doi.org/10.7554/eLife.37815.009

The following figure supplement is available for figure 3:

**Figure supplement 1.** Data processing methods.

DOI: https://doi.org/10.7554/eLife.37815.010

probability began to grow gradually as soon as the odor concentration started to decrease (*Figure 3D*, white arrow), but peaked close to the point where the linear off ramp returned to baseline (black arrow). This result suggests some form of sensitivity adaptation, that allows the fly to respond to a small decrease from a saturating concentration of odor. We also noted that upwind velocity remained positive during these ramps (*Figure 3C*, white arrow), suggesting that ON and OFF responses can be driven —at least partially— at the same time.

Finally, we wished to gauge the ability of flies to follow rapid fluctuations in odor concentration, as occurs in real odor plumes. Indeed, olfactory receptor neurons can follow odor fluctuations up to 10–20 Hz (*Nagel and Wilson, 2011*; *Kim et al., 2011*), and these rapid responses have been hypothesized to be critical for navigation in odor plumes (*Nagel and Wilson, 2011*; *Gorur-Shandilya et al., 2017*). To test the behavioral response of flies to rapid odor fluctuations, we used proportional valves to create ascending and descending frequency sweeps of 10% ACV between approximately 0.1 and 1 Hz (*Figure 3E–H*). The peak frequency we could present was limited to 1 Hz, as we found that frequencies higher than this became attenuated at the downwind end of the arena, presumably because odor diffuses as it is transported downwind, blurring the differences between peaks and troughs in the stimulus (see Materials and methods). In addition, we presented a 'plume walk': an odor waveform created by taking an upwind trajectory at fly pace through a boundary layer plume measured using planar laser imaging fluorescence (PLIF; *Figure 3I–J*, see Materials and methods, *Connor et al., 2018*).

As in previous experiments, we warped all behavioral data to account for the fact that flies encounter the odor fluctuations at different times depending on their position in the arena (*Figure 1—figure supplement 1* and Materials and methods). In addition, we excluded behavioral data points within 3 mm of the side walls, where boundary layer effects would cause slower propagation of the stimulus waveform. We also excluded responses occurring after each fly reached the upwind end of the arena, where arena geometry would constrain their direction of movement. The resulting traces represent our best estimate of the time courses of behavioral parameters (*Figure 3—figure supplement 1*), although we cannot completely rule out some contribution of odor diffusion or arena geometry.

We found that upwind velocity tracked odor fluctuations at the lowest frequencies, but that modulation became attenuated at higher frequencies (end of the ascending frequency sweep and start of the descending frequency sweep; *Figure 3E and G*), suggesting low-pass filtering of the odor signal. Similarly, upwind velocity peaked in response to nearly every fluctuation in the 'plume walk', but remained elevated during clusters of odor fluctuations (*Figure 3I*). The frequency-dependent attenuation was seen in both ascending and descending frequency sweeps, arguing against it being an effect of position in the arena, or duration of exposure to odor. Attenuation was not due to the filter imposed on trajectories during processing, as it was visible also when this filtering step was omitted (*Figure 3—figure supplement 1C–D*). We think it is also unlikely to be due to a limit on our ability to measure fast behavior reactions. We observed rapid decreases in ground speed in response to click stimuli that did not attenuate at higher frequencies (*Figure 3—figure supplement 1C,F*), arguing that the attenuation seen with odor does not reflect a limit on detecting rapid behavioral responses. Turn probability at offset showed even stronger evidence of low-pass filtering. Fluctuations in turn probability were attenuated during the higher frequencies of both frequency sweeps, and the strongest responses occurred at the end of the stimulus to the absence of odor (*Figure 3F, H,J*). The initial peaks in turn probability most likely represent the initial upwind turn, rather than an OFF response.

Together these experiments provide detailed measurements of the way that ON and OFF behaviors depend on the history of odor encounters. Moreover, they suggest that the two responses depend on odor history in different ways, with rapid fluctuations leading to elevated ON responses and suppressed OFF responses.

## Phenomenological models of ON and OFF responses

We next sought to develop computational models that could account for the behavioral dynamics described above. A challenge was that behavioral responses saturated at concentrations above 1% ACV, and they were also modulated by small decreases and fluctuations from a higher concentration (10%). This suggests some form of adaptation, in which the sensitivity of behavior to odorant shifts over time, allowing responses to occur near what was previously a saturating concentration. Sensitivity adaptation has been described at the level of olfactory receptor neuron transduction and can be implemented as a slow rightward shift in the Hill function that describes intensity encoding (*Kaissling et al., 1987*; *Nagel and Wilson, 2011*; *Gorur-Shandilya et al., 2017*). We therefore modeled adaptation by filtering the odor waveform with a long time constant $\tau_A$ and using the resulting signal to dynamically shift the midpoint of the Hill function to the right (see Materials and methods). The baseline $\kappa_d$ of the Hill function was taken from the fits in *Figure 3A and B*. We call this process 'adaptive compression' (*Figure 4A*) as it both compresses the dynamic range of the odor signal (from orders of magnitude to a linear scale), and adaptively moves the linear part of this function to the mean of the stimulus. We then tested four models for the ON response: one with adaptive compression followed by a low-pass filter ('ACF'), one with filtering followed by adaptive compression ('FAC'), and the same models without adaptation ('CF' and 'FC' respectively). We note that the FC model, with filtering followed by a fixed nonlinearity, is most similar to traditional linear-nonlinear models. For simplicity, we parameterized the low-pass filter by a single time constant $\tau_{ON}$, that describes the amount of smoothing seen in the response (Materials and methods).

We first fit models of the ON response to all upwind velocities shown in *Figure 3*, omitting and reserving the 'plume walk' stimulus to use as a test. We found that both models with adaptation performed better than models without, and that the model with adaptive compression first ('ACF', *Figure 4A*) outperformed the adaptive model with filtering first ('FAC', *Figure 4B*). As shown in *Figure 4C*, model ACF correctly predicted saturation with increasing odor concentration, and also the fact that responses to high odor concentrations exhibit adaptation while those to low odor concentrations do not. This model also correctly predicted the attenuation seen during frequency sweeps (*Figure 4D and E*), although some details of response timing early in the stimulus were not matched. We note that behavioral responses used for fitting were recorded in three different experiments with different sets of flies, and we used a single set of parameters to fit all responses; some differences between real and predicted response (for example the timing of response onset in *Figure 3D and E* vs C) may reflect differences in responses across experiments. The time constant of filtering was 0.72 s (see *Table 1*), significantly slower than encoding in peripheral ORNs (*Kim et al., 2011*; *Nagel and Wilson, 2011*). The time constant of adaptation was very slow (9.8 s). Models without adaptation (pink trace in *Figure 4D–E*) exhibited strong saturation during the frequency sweep, which was not observed experimentally.

We next fit the OFF response using four related models. In this case, the adaptive compression step was the same, but we used a differentiating filter instead of a low-pass filter, to generate responses when the odor concentration decreases from a previously high level. This filter was parameterized by two time constants, $\tau_{OFF1}$ and $\tau_{OFF2}$, that describe the time intervals over which the current and past odor concentrations are measured (*Figure 4F*, Materials and methods). Again we found that models with adaptation outperformed those without, and that the adaptive model with compression first very slightly outperformed the adaptive model with filtering first (*Figure 4G*). This model reproduced reasonably well the responses of flies to odor ramps (*Figure 4H*). The slow time constant of filtering was 4.84 s, accounting for the selectivity of the OFF response to low frequencies during frequency sweeps (*Figure 4I and J*). The time constant of adaptation was of similar magnitude to that derived from fitting the ON response (10.62 s).

To further assess the best-performing ON and OFF models (those with adaptive compression followed by filtering), we tested the performance of these models on the 'plume walk' stimulus. We found that the ON model reproduced most major contours in the 'plume walk' response (*Figure 4K*), although there was some discrepancy in the timing of peaks early in the response as for

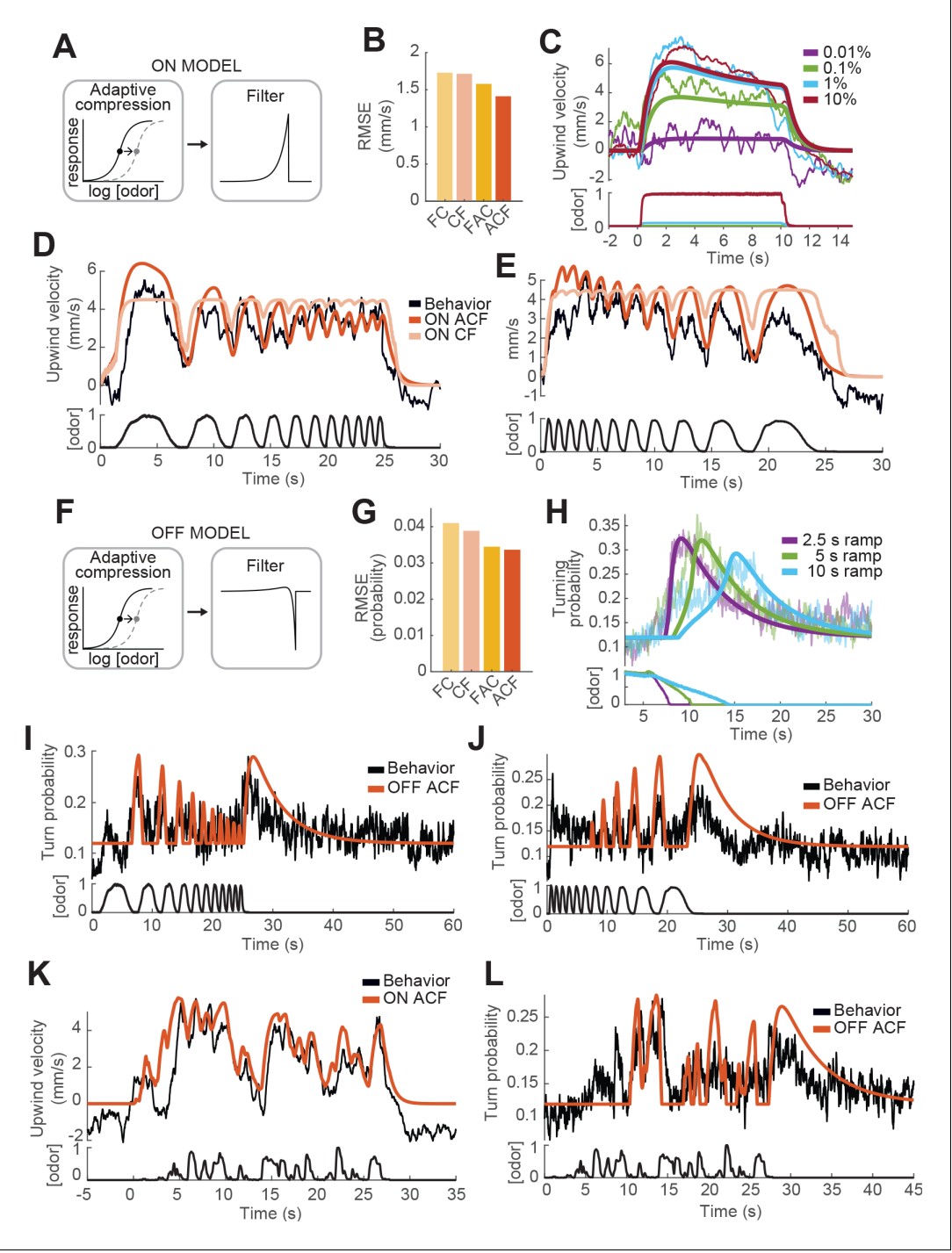

**Figure 4.** Computational modeling of ON and OFF response functions. (**A**) ON model schematic featuring adaptive compression followed by linear filtering. (**B**) Root mean squared error between predictions of four ON models and behavioral data. FC: filter then compress; CF: compress then filter; FAC: filter then adaptive compression; ACF: adaptive compression then filtering. (**C**) Upwind velocity of real flies (top thiner traces; average; same data in *Figure 3A*) and predictions of the ACF ON model (top thicker traces) to square pulses of ACV at different concentrations. Bottom traces: stimuli, normalized to maximal amplitude. Note that adaptation appears only at higher concentrations and that responses saturate between 1% and 10% ACV. (**D**) Upwind velocity of real flies (top black trace; average; same data in *Figure 3E*), and predictions of ACF (red) and CF (pink) ON models to an ascending frequency sweep. Bottom trace: stimulus. Note that the model without adaptation (CF) exhibits saturation not seen in the data. (**E**) Same as D for a descending frequency sweep stimulus (same data in

*Figure 4 continued on next page*

*Figure 4 continued*

*Figure 3G*). (F) OFF model schematic featuring adaptive compression followed by differential filtering. (G) Root mean squared error between predictions of four OFF models and behavioral data. (H) Turn probability of real flies (top thiner traces; average; same data in *Figure 3D*) and predictions of the ACF OFF model (top thicker traces) to odor ramps of different durations. Bottom traces: stimuli. (I) Turn probability (top black trace; average; same data in *Figure 3F*), and predictions of ACF OFF model (top red trace) to an ascending frequency sweep. Bottom trace: stimulus. (J) Same as I for a descending frequency sweep stimulus (same data in *Figure 3H*). (K) Upwind velocity (top black trace; average; same data in *Figure 3I*), and predictions of ACF ON model to the 'plume walk' stimulus (see Results). Bottom trace: stimulus. RMSE = 1.355. (L) Same as K for the same stimulus, showing turning probability of real flies (top black trace; average; same data in *Figure 3J*) and predictions of the ACF OFF model (top red trace). Bottom trace: stimulus. RMSE = 0.038. Plume walk responses were not used to fit the models.

DOI: https://doi.org/10.7554/eLife.37815.011

the frequency sweeps (*Figure 4D*). The OFF model also captured many of the major peaks in the behavioral response (*Figure 4L*), as well as the time course of the slow offset response after the end of the stimulus. Overall, the RMSE errors between predictions and data for the plume walks were comparable to those for the stimuli we used for fitting. We conclude that models featuring adaptive compression followed by linear filtering provide a good fit to behavioral dynamics over a wide range of stimuli.

## A model of olfactory navigation

To understand how the ON and OFF functions defined above might contribute to odor attraction, we incorporated our ON and OFF models into a simple model of navigation. In our model (*Figure 5A–C*), we propose that odor dynamics directly influence ground speed and turn probability through the ON and OFF functions developed and fit above. Specifically, $ON(t)$ drives an increase in ground speed and a decrease in turn rate, leading to straight trajectories, while $OFF(t)$ drives a decrease in ground speed and an increase in turn rate, leading to local search (*Figure 5B*). Ground speed ($v$) and turn probability ($P(t)$) of our model flies are then defined by

$$v(t) = v_0 + \kappa_1 ON(t) - \kappa_2 OFF(t) \tag{1}$$

**Table 1.** Values of ON and OFF functions parameters.

Results of fitting the different ON and OFF functions to behavioral data by non-linear regression. Highlighted in green are the models of choice and the parameters that were used in the navigation model and the simulations shown in *Figures 5* and *6*. $\tau_x$: different time constants of ON, OFF and adaptation filters. RMSE: root mean squared error between predictions of the models and the corresponding data they were fitted to. Corr.Coef.: Pearson's linear correlation coefficients between predictions of the models and the corresponding data they were fitted to.

| ON MODEL | $\tau_{ON}$ | — | $\tau_A$ | $scale_{ON}$ | RMSE | Corr.Coef. |
|---|---|---|---|---|---|---|
| Filtering then adaptive compression (FAC) | 0.34 | — | 20.36 | 5.9 | 1.5784 | 0.89 |
| Adaptive compression then filtering (ACF) | 0.72 | — | 9.8 | 7.3 | 1.4122 | 0.92 |
| Filtering then compression (FC) | 0.04 | — | — | 4.4 | 1.747 | 0.85 |
| Compression then filtering (CF) | 0.3 | — | — | 4.5 | 1.7058 | 0.86 |
| OFF MODEL | $\tau_{OFF1}$ | $\tau_{OFF2}$ | $\tau_A$ | $scale_{OFF}$ | RMSE | Corr.Coef. |
| Filtering then adaptive compression (FAC) | 0.76 | 3.96 | 16.7 | 0.3 | 0.0345 | 0.75 |
| Adaptive compression then filtering (ACF) | 0.62 | 4.84 | 10.08 | 0.6 | 0.0336 | 0.77 |
| Filtering then compression (FC) | 0.58 | 3 | — | 0.1 | 0.0409 | 0.62 |
| Compression then filtering (CF) | 0.06 | 5.02 | — | 0.3 | 0.0389 | 0.69 |

DOI: https://doi.org/10.7554/eLife.37815.012

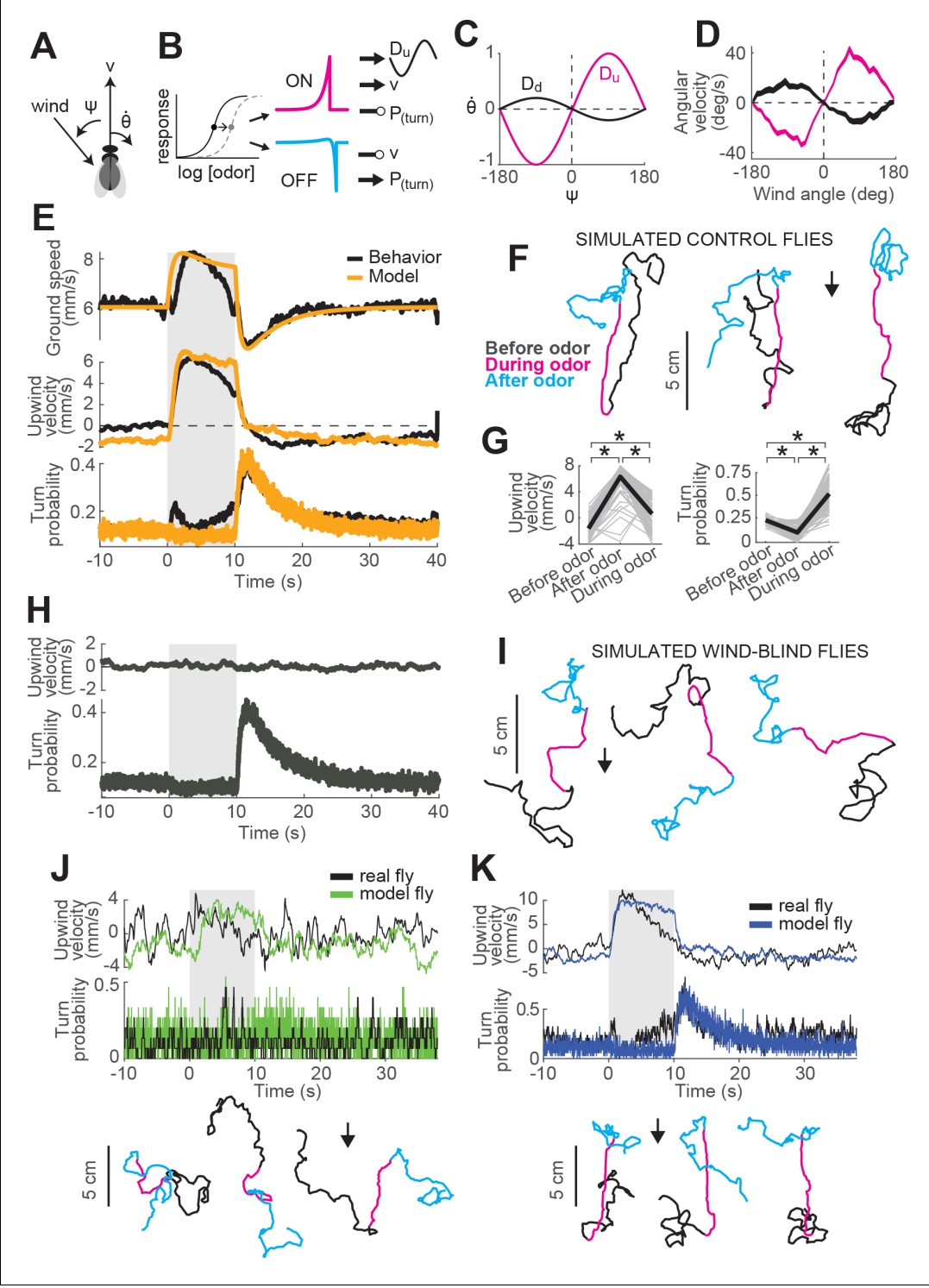

**Figure 5.** A navigation model based on ON and OFF functions can recapitulate many aspects of our behavioral data. (**A**) Schematic of a fly showing model outputs (v: ground speed; $\dot{\theta}$: angular velocity) and input ($\psi$: wind angle with respect to the fly). (**B**) Schematic of the model algorithm. Odor stimuli are first adaptively compressed, then filtered to produce ON (magenta) and OFF (cyan) functions. These functions modulate ground speed and angular velocity of the simulated fly. Angular velocity has both a stochastic component controlled through turn probability and a deterministic component guided by wind. (**C**) Wind direction influences behavior through two sinusoidal D-functions which drive upwind (magenta) and downwind (black) heading respectively. A weak downwind drive is always present, while a stronger upwind drive is gated by the ON function. (**D**) D-functions (average angular

*Figure 5 continued on next page*

*Figure 5 continued*

velocity as a function of wind angle with respect to the fly) calculated from responses of real flies (data from *Figure 1*, mean±SEM, n = 75 flies, 1306 trials). Magenta trace: data from 0 to 2 s during odor. Black trace: 0–2 s after odor. (**E–G**) Simulated trajectories of model flies are similar to those of real flies. (**E**) Ground speed, upwind velocity and turn probability (average; n = 75 flies, 1306 trials) from real flies (black; data from *Figure 1*) and from 500 trials simulated with our model (orange) in response to a 10 s odor pulse. (**F**) Example trajectories from the simulation in E. Black: before odor. Magenta: during odor. Cyan: after odor. Black arrow: direction of the wind. (**G**) Mean values of upwind velocity and turn probability from the model simulations in E, before (−30 to 0 s), during (2 to 3 s) and after (11 to 13 s) the odor pulse. Gray lines: data from individual trials. Black lines: group average. Horizontal lines with asterisk: Statistically significant changes in a Wilcoxon signed rank paired test after correction for multiple comparisons using the Bonferroni method (see Materials and methods for p values). n.s.: not significant. (**H–I**) Simulated trajectories of wind-blind flies. (**H**) Upwind velocity and turn probability (average) from 500 trials simulated in response to a 10 s odor pulse with no wind (both D-functions coefficients set to 0) to mimic the responses of wind-blind flies (see *Figure 2*). Note the absence of modulation in upwind velocity. (**I**) Example trajectories from the simulation in H. Color code and arrow as in F. Note that trajectories preserve the characteristic shapes of the ON and OFF responses but lack any clear orientation during ON responses. (**J–K**) Simulated trajectories of weak and strong-searching flies. (**J**) Upwind velocity and turn probability of one weak-searching fly. Real fly appears in green-highlighted examples in *Figure 1—figure supplement 2* (here black traces; average; n = 15 trials). The model simulation (green traces; average; n = 15 trials) was created by using the mean upwind velocity and turn probability for this fly (*Figure 1—figure supplement 2*, green) as a fraction of the population average upwind velocity and turn probability to scale the ON and OFF functions (values used: ON scale = 0.3, OFF scale = 0.26). Bottom: example trajectories from the model simulation, compare directly to *Figure 1—figure supplement 2A* left (color code and arrow as in F). (**K**) Equivalent to J, for one strong-searching fly (n = 34 trials). Compare blue-highlighted examples in *Figure 1—figure supplement 2* with the model simulation (n = 34 trials; values used: ON scale = 1.9, OFF scale = 1.6).

DOI: https://doi.org/10.7554/eLife.37815.013

$$P(t) = P_0 - \kappa_3 ON(t) + \kappa_4 OFF(t) \tag{2}$$

where $v_0$ and $P_0$ are baseline values extracted from behaving flies (*Figure 1F*).

Second, we propose that turning has both a probabilistic component, driven by odor, and a deterministic component, driven by wind. In the absence of any additional information about how these turn signals might be combined, we propose that they are simply summed. To model deterministic wind-guided turns, we constructed a sinusoidal desirability function or 'D-function' which drives right or leftward turning based on the current angle of the wind with respect to the fly. Such functions were originally proposed to explain orientation to visual stripes (*Reichardt and Poggio, 1976*). In an upwind D-function, wind on the left (denoted by negative $\psi$ values) drives turns to the left (denoted by negative $\dot{\theta}$ values), and vice-versa (*Figure 5C*, magenta trace). Conversely, in a downwind D-function, wind on the left drives turns to the right, and vice-versa (black trace). Supporting the notion of a wind direction-based D-function, we found that the average angular velocity as a function of wind direction in the period immediately after odor onset had a strong 'upwind' shape (*Figure 5D*, magenta trace), while the angular velocity after odor offset had a weaker 'downwind' shape (*Figure 5D*, black trace). In our navigation model, the angular velocity of the fly is then given by

$$\dot{\theta}(t) = \rho(t)G + \kappa_5 ON(t)D_u(\varphi) + \kappa_6 D_d(\varphi) \tag{3}$$

where $\rho(t)$ is a binary Poisson variable with rate $P(t)$ and $G$ is the distribution of angular velocities drawn from when $\rho$ is 1 (see Materials and methods). This first term generates probabilistic turns whose rate depends on recent odor dynamics. The second term is an upwind D-function, gated by the ON function, that produces strong upwind orientation in the presence of odor. The final term is a constant weak downwind D-function that produces a downwind bias in the absence of odor.

This navigation model is parameterized by six coefficients ($\kappa_1$-$\kappa_6$) that determine the strength with which the ON and OFF functions modulate ground speed, turn probability, and the drive to turn up- or downwind. For example, $\kappa_1$ determines how much the forward velocity increases when the ON function increases by a specific amount. We first adjusted these parameters so that average motor

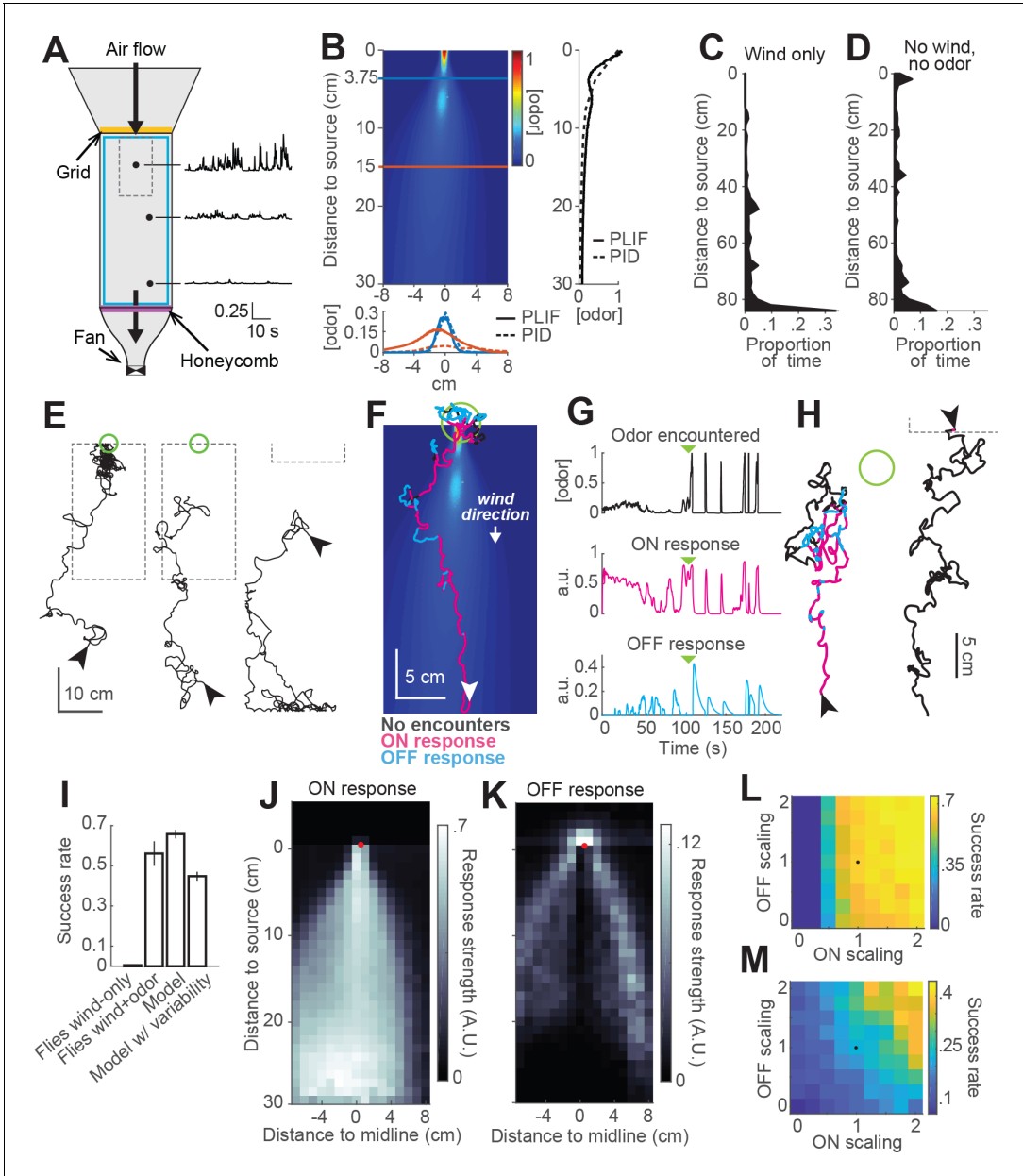

**Figure 6.** Real and virtual behavior of flies in a turbulent odor environment. (**A**) Schematic of a turbulent wind tunnel used for behavioral experiments and PLIF imaging (top view; see Materials and methods). Black arrows: direction of air flow drawn by fan at downwind end; top arrow coincides with the tube carrying odor to the arena. Black dots and associated traces: sites of PID measurements (and corresponding signals; units normalized to mean concentration near the odor source). Smaller dashed square: Area covered with the PLIF measurements in the Colorado wind-tunnel (see Materials and methods). Yellow line: position of the wooden dowel grid. Purple line: position of the honeycomb filter. Blue square: perimeter moat filled with water. (**B**) PLIF measurements of an odor plume (average of 4 min of data). Blue/red horizontal lines: Sites of cross-sections (bottom plot). Bottom plot: cross-sections of the plume measured with PLIF (solid lines; 4 min average) and PID (dashed lines; 3 min average). Right plot: Odor concentration along midline of the plume (x = 0) measured with PLIF and PID (4 and 3 min average, respectively). All measurements in B appear normalized to average odor concentration at the source. (**C–D**) Flies exhibit a downwind preference in the turbulent wind tunnel. (**C**) Distribution of fly positions during trials with wind but no odor (n = 14 flies/trials). (**D**) Same as C, during trials with no wind (n = 13 flies/trials). (**E**) Example trajectories of flies during trials with an odor plume. From left to right: a successful trial in which the fly came within 2 cm of the source; intermediate trial in which the fly searched but did not find the source; failed trial where fly moved downwind. Arrowheads: starting positions. Green circles: 2 cm area around odor source. Dashed gray lines: area covered by PLIF measurements (use as positional reference; right-most trace shows only lower section of outline). (**F**) Example trajectory of a model fly that successfully found the odor source (background image from B). Colors show times when ON > 0.1 (magenta) or OFF > 0.05 (cyan). White arrowhead: Starting position and orientation. Green circle: 2 cm area around source. (**G**) Time courses of odor concentration encountered along the trajectory in F, with corresponding ON and OFF responses. Green arrowheads: time of entrance into the green circle. (**H**) Example trajectories of model

*Figure 6 continued on next page*

*Figure 6 continued*

flies (color code, green circle and arrowheads as in F). Left trace and green circle associated: intermediate trial where fly searched but did not find the source. Right trace: failed trial where fly moved downwind. Dashed line: lower section of the plume area. (I) Performance (proportion of successful trials±SE; see Materials and methods) of real and model flies in a plume. Data from real flies on trials with only wind (n = 13 flies/trials) and trials with wind and odor (n = 14 flies/trials). Model data using parameters fit to the mean fly in every trial (n = 500 trials; see Results). Model with variable ON and OFF scaling, reflecting variability in ON/OFF responses across individuals (n = 500 trials; see Results and *Figure 1—figure supplement 2*). (J–K) Average strength of ON (J) and OFF (K) responses as a function of position for model flies in the plume (data from simulation with mean parameters). Red dots: odor source. Note that ON is high throughout the odor plume, especially along its center, while OFF is highest at the plume edges. (L) Performance of the model in a plume (proportion of successful trials) with different scaling factors applied to ON and OFF responses. Black dot: performance of model using fitted values. (M) Same as L for model flies navigating a simulated odor gradient with a gaussian distribution and no wind (see Materials and methods).

DOI: https://doi.org/10.7554/eLife.37815.015

parameters calculated from simulations of our model in response to a 10 s odor pulse would match the ground speed, upwind velocity, and turn probability of the 'mean fly' seen in *Figure 1* (*Figure 5E*, see Materials and methods and *Table 2*). Similar to real flies, this model produced upwind runs during the odor pulse and searching after odor offset (*Figure 5F*). Average upwind velocity during the odor and turn probability after the odor were comparable to measurements from real flies (compare *Figure 5G* and *Figure 1G*). As a second test, we set the coefficients controlling wind orientation ($\kappa_5$ and $\kappa_6$) to zero, making the model fly indifferent to wind direction and mimicking a wind-blind real fly. In this case, the model produced undirected runs during odor and search behavior at odor offset, as in our data (compare *Figure 5H–I* and *Figure 2A–B*).

We also asked whether our model could account for variability in behavior seen across flies (*Figure 1—figure supplement 2*). To address this question, we asked whether differences in behavior could be accounted for by applying fly-specific scale factors to the ON and OFF functions of the model. To define these scale factors, we returned to our main data set (*Figure 1*) and computed an ON scale value for each fly equal to its mean upwind velocity, divided by the mean upwind velocity across flies. An OFF scale value was computed similarly by taking the mean turn probability for a fly divided by the mean across flies. This procedure allowed us to express the behavior of each fly as a scaled version of the group average response. Next, keeping all other parameters in our navigation model fixed as previously fitted, we scaled the ON and OFF functions to match the value of individual flies. The trajectories produced by these scaled models resembled the behavior of individual flies both qualitatively and quantitatively. For example, scaling down the ON and OFF functions produced similar behavior to a weak searching fly (*Figure 5J*, compare directly to green-highlighted examples in *Figure 1—figure supplement 2A*), while scaling up the ON and OFF function produced

**Table 2.** Values of navigation model parameters used in all simulations in this article, with their function in the model explained.

**Navigation model**

| Parameter | Value | Units | Role |
|---|---|---|---|
| $P_0$ | 0.12 | Rate | Baseline turn rate |
| $\sigma$ | 20 | deg/s | Standard deviation of angular velocity distribution |
| $v_0$ | 6 | mm/s | Baseline ground speed |
| $\kappa_1$ | 0.45 | mm/s | Strength of ON speed modulation |
| $\kappa_2$ | 0.8 | mm/s | Strength of OFF speed modulation |
| $\kappa_3$ | 0.03 | — | Strength of ON turning modulation |
| $\kappa_4$ | 0.75 | — | Strength of OFF turning modulation |
| $\kappa_5$ | 5 | deg/sample | Strength of ON upwind-drive modulation |
| $\kappa_6$ | 0.5 | deg/sample | Strength of downwind-drive modulation |

DOI: https://doi.org/10.7554/eLife.37815.014

behavior similar to a strongly-searching fly (**Figure 5K**, compare directly to blue-highlighted examples in **Figure 1—figure supplement 2A**).

Together, these results support the idea that our model captures essential features of how flies respond to odor and wind in miniature wind-tunnels, including the responses of intact and wind-blind flies, and variations in behavior across individuals. Thus, this model provides a basis for examining the predicted behavior of flies in more complex environments.

## Behavior of real and model flies in a turbulent environment

Finally, we sought to test whether our model could provide insight into the behavior of real flies in more complex odor environments. To that end we constructed two equivalent wind tunnels capable of delivering a turbulent odor plume (**Figure 6A**; see Materials and methods). In one tunnel (New York), we incorporated IR lighting below the bed and cameras above it to image fly behavior in response to a turbulent odor plume. In the second tunnel (Colorado), we used a UV laser light sheet and acetone vapor to obtain to high-resolution movies of the plume for use in modeling (**Figure 6B**,

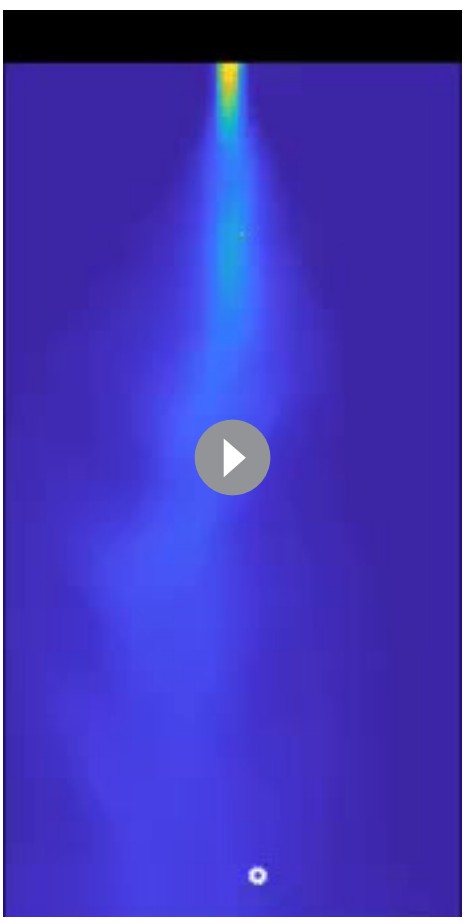

**Connor et al., 2018**). These two apparatuses had similar dimensions, and matched odor delivery systems and wind speeds. We used photo-ionization detector measurements to corroborate that the shape and dynamics of the plume in the New York tunnel was similar to the one measured in Colorado (**Figure 6B**).

We next examined the behavior of walking flies in this wind tunnel. Flies were of the same genotype and were prepared for experiments in the same way as those used previously. They were constrained to walk by gluing their wings to their backs with a small drop of UV glue and by placing a 1cm-wide water-filled moat at the edge of the arena.

We first tested flies with wind only (no odor) at 10 cm/s. As in our miniature wind tunnels, we found that flies uniformly preferred the down-wind end of the arena (**Figure 6C**). In the absence of wind, this preference was reduced (**Figure 6D**). We observed no preference for the upwind end of the tunnel (which received greater ambient light from the room) or for the odor tube, confirming that these *norpA*[36] flies lacked phototaxis and visual object attraction. Finally, we examined behavior in the presence of a plume of ACV 10%, and we observed diverse responses (**Figure 6E**). Of 66 flies, 37 (56%) successfully located the odor source, walking upwind and lingering in a small region close to the odor tube (**Figure 6E**, left trace). Other flies searched in the middle of the arena without getting close to the source (**Figure 6E**, middle trace, 18%), while others headed downwind and remained at the downwind end of the arena (**Figure 6E**, right trace, 15%). The rest of the flies (seven flies) either moved very little or moved mostly along the sides of the tunnel.

To compare the performance of our model to the behavior of the flies, we ran simulations with our model using the plume movie measured in the Colorado wind tunnel as a virtual

**Video 2.** Behavior of a model fly navigating an odor plume. The video shows 3 min long trial, sped up four times. The background image represents the odor concentration of the plume (equivalent to **Figure 6B**) recorded by PLIF in the Colorado wind tunnel (see Materials and methods). The moving dot represents the position of the model fly, with changing colors depending on its current behavior. Magenta dot: ON response is larger than 0.1. Cyan dot: OFF response is larger than 0.05. White circle: no odor-evoked responses.
DOI: https://doi.org/10.7554/eLife.37815.016

environment (*Video 2*). At each time step, we took the odor concentration at the location of the simulated fly and used this to iteratively compute ON and OFF functions and update the fly's position accordingly (*Figure 6F–H*). We observed that model flies produced trajectories similar to those of real flies in the wind tunnel. For example, some flies responded to odor with general movement upwind interrupted by occasional excursions out of the plume (*Figure 6F*); overall, 66% successfully came within 2 cm of the odor source. Other model flies searched but failed to locate the source (17% of trials; *Figure 6H*, left trace), while others 'missed' the plume and moved downwind (17% of trials; *Figure 6H*, right trace). Using a single set of model parameters fit to the mean behavioral responses in *Figure 1F*, we found that our model yielded a similar —although somewhat higher— success rate than real flies (*Figure 6I*, 66% versus 56% success rate).

Given the large degree of variability in behavior across individuals, we wondered if this variability could account for the difference in success rates between real and model flies. We therefore ran simulations incorporating variability in fly behavior. In each trial of this simulation, we randomly drew a pair of ON and OFF scale values (as described previously) and used it to scale the ON and OFF responses of the model for that trial. Introducing variability in the model decreased the success rate to 45% (*Figure 6I*), and made it slightly worse than that of real flies in the wind tunnel. This simulation produced 27% 'failed' searches and 28% trials in which flies 'missed' the plume and went downwind.

The simulations described above indicate that the trajectories produced by our model in a turbulent environment are qualitatively similar to those produced by real flies. To gain insight into the roles that ON and OFF behaviors play in this environment, we color-coded model trajectories according to the magnitude of the ON and OFF functions underlying them (*Figure 6F–G*). We observed that the ON function was dominant throughout most of the odorized region, while excursions from the plume elicited strong OFF responses that frequently resulted in the model fly re-entering the plume. OFF responses were also prominent near the odor source, where they contributed to the model fly lingering as observed in real flies. ON and OFF magnitudes varied over a much smaller range than the range of odor concentrations, suggesting that the adaptive compression we incorporated into the model helps flies to respond behaviorally over a greater distance downwind of the source. Plotting the strengths of both responses as a function of position in an odor plume supported this analysis of individual trajectories (*Figure 6J–K*). This analysis showed ON being active in the area within the plume, and more active the closer to the center of the plume (*Figure 6J*), where the concentration of odor is higher and intermittency is lower. This suggests that ON responses are responsible for making flies progress within the odor area, allowing them to eventually reach the odor source. The OFF function was most active in the area surrounding the odor plume (*Figure 6K*), suggesting it plays a role in relocating the plume after flies walk outside of it and the odor signal is lost. OFF values were also high just upwind of the source. Notably, OFF values were generally low within the plume, even though large fluctuations do occur within this region. This suggests that the slow integration time of the OFF response may help it to detect the edges of the time-averaged plume, allowing flies to slow down and search only when the plume has genuinely been exited.

To assess the relative role of ON and OFF functions in promoting source localization, we ran a series of simulations in an odor plume (500 trials each), systematically changing the scaling factors of the ON and OFF functions (*Figure 6L*). We observed that performance increased with both functions, but that ON was more critical for success in the plume, producing large improvements in performance as it increased. This is consistent with the idea that wind direction is a highly reliable cue in this environment (indeed, it is likely more reliable in our model than in reality, as we did not incorporate local variations in flow induced by turbulence into our model). To test the idea that ON and OFF might have different importance in a windless environment, we repeated the analysis just described in a simulated Gaussian odor gradient with no wind (*Figure 6M*). In this environment, success rates were lower, but the contributions of ON and OFF were more similar, with higher success rates when the OFF function was the strongest for any given strength of the ON function. These results suggest that ON and OFF responses have different impact on success depending on the features of the environment.

## Role of spatial comparisons in plume navigation

In addition to the ON and OFF functions described here, walking *Drosophila* have also been shown to perform spatial comparisons across their antennae, and to turn toward the antenna that receives a higher odor concentration (*Borst and Heisenberg, 1982*; *Gaudry et al., 2013*). Such turns can be produced using optogenetic activation of olfactory receptor neurons in one antenna, arguing that they are independent of wind sensing (*Gaudry et al., 2013*). Because the fly's antennae are located so close to one another, and because it has been unclear what kind of spatial information a plume provides, the role of these spatial comparisons in plume navigation has been questioned (*Borst and Heisenberg, 1982*). To ask whether such comparisons could contribute to source finding in the boundary layer plume that we measured, we incorporated a fourth term into the total angular velocity in our model:

$$\dot{\theta}(t) = \rho(t)G + \kappa_5 ON(t)D_u(\varphi) + \kappa_6 D_d(\varphi) + \kappa_7(C_l - C_r) \tag{4}$$

Here, $C_l$ and $C_r$ represent the odor concentrations at the left and right antennae, processed by the same adaptive compression function used previously (see Materials and methods). The left antenna was taken to be at the position of the fly, and the right antenna was taken to be one pixel (740 $\mu$m) to the right. The results of these simulations depended heavily on the choice of gain $\kappa_7$. Based on the results of (*Borst and Heisenberg, 1982*) and (*Gaudry et al., 2013*), we estimated a gain of approximately 40 deg/s when the concentration difference between the two antennae is maximal. In this case, spatial comparisons did not contribute significantly to the probability of successfully finding the source (*Figure 7A–C*). However, if we increased the gain to 300 deg/s, we found that performance of the model improved significantly, from 67% to 76%. Under these conditions, trajectories remained closer to the center of the plume and were less dispersed around the source (*Figure 7A–B*, third column). We observed a contrary phenomenon when we switched the position of the antennae in the model, so that information from the right side was interpreted as left, and vice-versa. This made model flies more prone to leave the area of the plume and wander off, decreasing their success rate to 54% (*Figure 7A–C*, fourth column). In the absence of wind sensation, flies performing a correct bilateral comparison were unable to locate the odor source (*Figure 7A–C*, fifth column). These results argue that nearby locations in the plume contain information that can be used to aid navigation (if the gain is high enough), but that this information is insufficient to find the odor source in the absence of wind.

To explore how performance depended on the interaction of wind sensation and spatial sensing, we varied the strength of these two behavioral components (*Figure 7D*). This analysis showed that some wind sensing is absolutely required to find the odor source, as almost no flies find the source when the wind coefficients are set to zero. However, in the presence of wind, bilateral sensing, controlled by $\kappa_7$, improves performance, with the greatest improvements coming at the highest gain. Thus, although the contributions of wind sensing and bilateral sensing sum linearly to control angular velocity in our model, their effects on finding the source are nonlinear, presumably because of the structure of the plume itself.

In addition, we asked whether both temporal sensing and spatial sensing contribute to performance in the plume. To do this, we varied the magnitude of the OFF response and the gain of bilateral sensing (*Figure 7E*), while keeping the strength of wind sensation constant. In this case, we observed that both components contributed to increased performance. This is consistent with our observations of model trajectories, which suggest that the OFF response and bilateral sensing work together to help reorient model flies into the plume when they wander out of it.

Together these results suggest that three different forms of sensation—flow sensing (wind), temporal sensing (OFF response), and spatial sensing (bilateral comparisons)—can all contribute to finding an odor source, but that the precise contribution of each mechanism depends both on the environment and on the gain or sensitivity of the animal to each measurement. These data support the idea that olfactory navigation in complex environments can be decomposed into several largely independent sensori-motor transformations and provide a foundation for investigating the neural basis of these components.

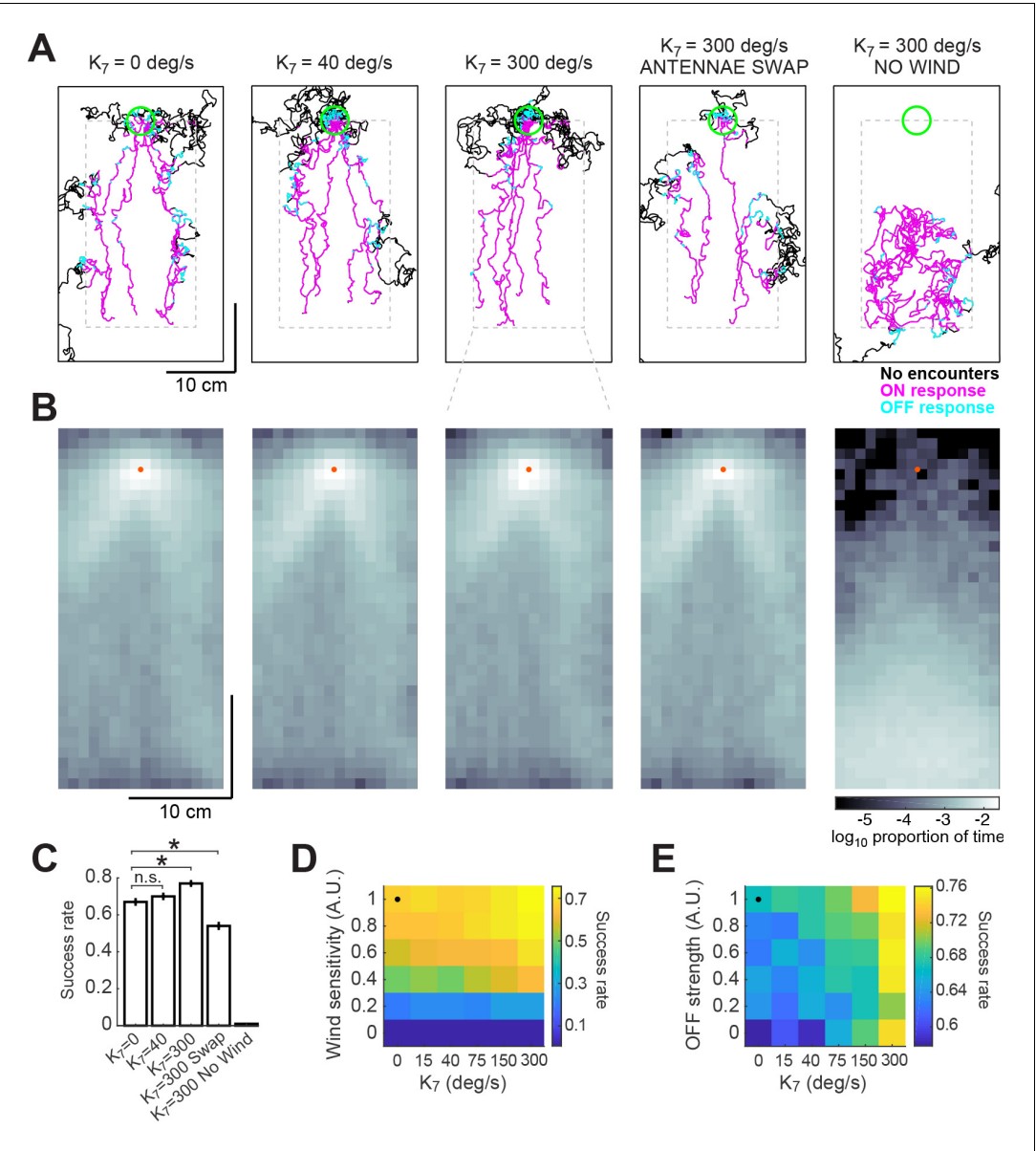

**Figure 7.** Addition of a bilateral sampling component can improve olfactory navigation. (**A**) Example trajectories from a series of model simulations of 500 trials each. In the first simulation the model was unchanged (as in *Figure 6*). In the second and third simulations, a bilateral component was added to total angular velocity with gain values ($\kappa_7$) of 40 and 300 deg/s, respectively. In the fourth simulation, all components of the model were active, but the information from the antennae was swapped —left was interpreted as right, and vice-versa. In the fifth simulation, wind sensation was turned off. Trajectories' colors show times when ON > 0.1 (magenta) or OFF >0.05 (cyan). Dashed gray lines: area of odor plume data (outside this area odor concentration is zero). A larger area is shown to display the behavior more clearly. Green circle: area of 2 cm around the odor source, used to define success in trials. (**B**) Density maps of flies' positions (logarithm of the proportion of total time) corresponding to each of the simulations in A, with data only from the areas within the dashed lines in A. Orange dots: position of the odor source. (**C**) Success rate (proportion of successful trials) in each of the simulations in A (average±SEM; see Materials and methods). Horizontal lines with asterisk: Statistically significant changes (see Materials and methods for details and p values). n.s.: not significant. (**D**) Performance of the model in a plume (sucess rate) as a function of wind sensitivity and strength of the bilateral component. Note that values for $\kappa_7$ don't scale linearly. Black dot: performance of model using fitted values (see Results). (**E**) Equivalent to D, showing performance as a function of strength of the OFF response and of the bilateral component.

DOI: https://doi.org/10.7554/eLife.37815.017

## Discussion

### Quantitative measurement of olfactory attraction behavior in adult fruit-flies

The ability to navigate toward attractive odors is widespread throughout the animal kingdom and is critical for locating both food and mates (*Bell and Tobin, 1982*). Taxis toward attractive odors is found even in organisms without brains, such as *E. coli*, and is achieved by using activation of a receptor complex to control the rate of random re-orientation events, called tumbles or twiddles (*Falke et al., 1997*). Precise quantification of the behavior elicited by controlled chemical stimuli has been critical to the dissection of neural circuits underlying navigation in gradient navigators such as *C. elegans* (*Gray et al., 2005*) and *Drosophila* larvae (*Tastekin et al., 2015*).

Larger organisms that navigate in air or water face fundamentally different problems in locating odor sources (*Cardé and Willis, 2008*; *Murlis et al., 1992*). Odors in open air are turbulent. Within a plume, odor concentration at a single location fluctuates over time, and local concentration gradients often do not point toward the odor source (*Crimaldi and Koseff, 2001*; *Webster and Weissburg, 2001*). To solve the problem of navigating in turbulence, many organisms have evolved strategies of combining odor information with flow information. For example, flying moths and flies orient upwind using optic flow cues during odor (*Kennedy and Marsh, 1974*; *David et al., 1983*; *van Breugel and Dickinson, 2014*). Marine invertebrates travel upstream when encountering an attractive odor (*Page et al., 2011*). Although neurons that carry signals appropriate for guiding these behaviors have been identified (*Olberg, 1983*; *Namiki et al., 2014*), a circuit-level understanding of these behaviors has been lacking. Obtaining such an understanding will require quantitative measurements of behavior coupled with techniques to precisely activate and inactivate populations of neurons.

In recent years, the fruit-fly *D. melanogaster* has emerged as a leading model for neural circuit dissection (*Simpson, 2016*). The widespread availability of neuron-specific driver lines, the ease of expressing optogenetic reagents, and the ability to perform experiments in a high-throughput manner have established the fruit-fly as a compelling experimental model. Here, we have developed a high-throughput behavioral paradigm for adult flies that allows for precise quantification of fly movement parameters as a function of well-controlled dynamic odor and wind stimuli. An important distinction between our paradigm, and others previously developed for flies (*Jung et al., 2015*; *van Breugel and Dickinson, 2014*; *Bell and Wilson, 2016*), is that it allows us to control the odor and wind stimuli experienced by the flies regardless of their movement. This 'open loop' stimulus presentation allowed us to measure the dependence of specific behaviors on odor dynamics and history. In addition, our paradigm allows for movement in two dimensions (in contrast to *Steck et al., 2012*; *Bell and Wilson, 2016*), which allowed us to observe and quantify search behavior elicited by odor offset. By combining this paradigm with techniques to activate and silence particular groups of neurons, it should be possible to dissect the circuits underlying these complex multi-modal forms of olfactory navigation.

### Unimodal and multimodal responses guide olfactory navigation in adult *Drosophila*

In our behavioral paradigm, we observed two distinct behavioral responses to a pulse of apple cider vinegar: an upwind run during odor, and a local search at odor offset. Previous studies have suggested that flies cannot navigate toward odor in the absence of wind (*Bell and Wilson, 2016*), while others have suggested that odor modulates multiple parameters of locomotion, resulting in an emergent attraction to odorized regions (*Jung et al., 2015*). Our findings suggest a synthesis of these two views. We find that upwind orientation requires wind cues transduced by antennal mechanoreceptors. In contrast, offset searching is driven purely by changes in odor concentration. In computational model simulations, we found that when wind provided a reliable cue about source direction, wind orientation was the major factor in the success of a model fly in finding the source. However, when wind cues were absent, ON and OFF behaviors both played equal roles. In real environments, wind direction is rarely completely reliable (*Murlis et al., 2000*), so both behaviors are likely to contribute to successful attraction.

The ON and OFF responses that we describe here have clear correlates in behaviors described in other organisms. The upwind run during odor has been described previously (*Flügge, 1934*; *Steck et al., 2012*) and seems to play a similar role to the upwind surge seen in flying insects (*Vickers and Baker, 1994*). Upwind orientation in walking flies appears to depend entirely on mechanical cues while upwind orientation during flight has been shown to be sensitive to visual cues (*Kennedy, 1940*; *Kennedy et al., 1981*; *van Breugel and Dickinson, 2014*). Searching responses after odor offset have been observed in walking cockroaches (*Willis et al., 2008*), and have been observed in adult flies following removal from food (*Dethier, 1976*; *Kim and Dickinson, 2017*) but have until recently not been reported in flies in response to odor (*Sayin et al., 2018*). The OFF response seems to play a role related to casting in flying insects, allowing the fly to relocate an odor plume once it has been lost, although the response we observed did not have any component of orientation orthogonal to the wind direction, as has been described in flight (*Kennedy and Marsh, 1974*; *van Breugel and Dickinson, 2014*). OFF responses were weaker in flies lacking the *norpA*[36] allele, suggesting that vision may be able to substitute to some degree for search behavior, or that the *norpA*[36] allele itself promotes more vigorous searching.

## Temporal features of odor driving ON and OFF behaviors

A common feature of chemotaxis strategies across organisms is the use of temporal cues to guide behavior. In gradient navigators, the dependence of behavior on temporal features of odor is well established. Bacteria respond to decreases in attractants over an interval of about 2 s (*Block et al., 1982*). Pirouettes in *C. elegans* are driven by decreases in odor concentration over a window of 4–10 s (*Pierce-Shimomura et al., 1999*). The temporal features of odor that drive behavioral reactions in plume navigators are less clear. Studies of moth flight trajectories in a wind tunnel have suggested that moths respond to each filament of odor with a surge and cast (*Baker, 1990*; *Vickers and Baker, 1994*), and cease upwind flight in a continuous miasma of odor (*Kennedy et al., 1981*). These findings have led to the idea that the rapid fluctuations found in plume are critical for promoting upwind progress (*Baker, 1990*; *Mafra-Neto and Cardé, 1994*). In contrast, *Drosophila* have been observed to fly upwind in a continuous odor stream (*Budick and Dickinson, 2006*), suggesting that a fluctuating stimulus is not required to drive behavior in this species. Flight responses to odor have been described as fixed reflexes (*van Breugel and Dickinson, 2014*), although they have also been shown to depend on odor intensity and history (*Pang et al., 2018*). Measurement of these dependencies has been hampered by the inability to precisely control the stimulus encountered by behaving animals.

Here, we have used an open loop stimulus and a very large number of behavioral trials, to directly measure the dependence of odor-evoked behaviors on odor dynamics and history. We find that in walking *Drosophila*, ON behavior (upwind orientation) is continuously produced in the presence of odor. ON behavior exhibited a filter time constant of 0.72 s, significantly slower than encoding of odor by peripheral olfactory receptor neurons (*Kim et al., 2011*; *Nagel and Wilson, 2011*). We think it is unlikely that this represents a limit on our ability to measure behavioral reactions with high temporal fidelity, as we observed very rapid, short-latency freezing in response to valve clicks that were faster and more reliable than olfactory responses. One possible explanation for this difference is that olfactory information may be propagated through multiple synapses before driving changes in motor behavior, while the observed freezing may be a reflex, executed through a more direct coupling of mechanoreceptors and motor neurons.

OFF responses (increases in turn probability) were driven by differences between the current odor concentration, and an integrated odor history with a time constant of 4.8 s. This long integration time was evident in responses to frequency sweeps and to the 'plume walk', where increases in turn probability were only observed in response to relatively slow odor fluctuations, or to long pauses between clusters of odor peaks. This filtering mechanism may allow the fly to ignore turbulent fluctuations occurring within the plume, and to respond with search behavior only when the overall envelope of the plume is lost. The neural locus of this offset computation is unclear. Olfactory receptor neurons that are inhibited in the presence of odor can produce offset responses when odor is removed (*Nagel and Wilson, 2011*); such inhibitory responses are generally odorant specific (*Hallem and Carlson, 2006*). In addition, inhibition after odor offset is observed in many olfactory receptor neurons, and the dynamics of this inhibition have been shown to predict offset turning in Drosophila larvae (*Schulze et al., 2015*). Alternatively, the OFF response could be computed

centrally in the brain. For example, many local interneurons of the antennal lobe are broadly inhibited by odors (*Chou et al., 2010*) and exhibit offset responses driven by post-inhibitory rebound (*Nagel and Wilson, 2016*). Rebound responses grow with the duration of inhibitory current (*Nagel and Wilson, 2016*), providing a potential mechanism for slow integration. Experiments testing the odor and glomerulus specificity of the OFF response could be used to distinguish between these possibilities, as ORN temporal responses are specific to particular odorants (*Hallem and Carlson, 2006*), while LN temporal responses are similar across odorants (*Chou et al., 2010*).

In addition to low-pass filtering, we found that behavioral responses to odor were best fit by models that included a compressive nonlinearity—in the form of a Hill function—whose sensitivity was slowly adjusted by adaptation. This type of adaptive compression has been observed in the transduction responses of *Drosophila* olfactory receptor neurons (*Kaissling et al., 1987*; *Nagel and Wilson, 2011*; *Gorur-Shandilya et al., 2017*). Additional adaptation has been observed at synapses between first and second order olfactory neurons (*Nagel et al., 2015*; *Cafaro, 2016*). Adaptation at multiple sites in the brain may contribute to the relatively slow adaptation time constants we measured for behavior (9.8 and 10 s for ON and OFF respectively.) Our adaptive compression model has some similarity to the quasi-steady state model of (*Schulze et al., 2015*), in which sensitivity to odor is dynamically adjusted to a running average of recent changes in odor history. Similar to that study in larvae, our study also suggests that events early in olfactory transduction can shape the time course of subsequent motor responses.

Why might olfactory behavior in walking flies reflect integration of olfactory information over time while upwind flight in moths appears to require a rapidly fluctuating stimulus? Several possibilities are worth considering. One is that the temporal demands of walking differ from those of flight. A flying moth travels at much faster speeds and over longer distances than a walking fly and will therefore traverse a plume in less time. Second, plumes developing near a boundary are broad and relatively continuous, while those in open air, particularly at the long distances covered by moths, are much more intermittent (*Crimaldi and Koseff, 2001*; *Celani et al., 2014*; *Yee et al., 1993*), again making detection of the plume edge potentially more important than responding rapidly to each plume encounter. Finally, receptor-odorant interactions can have different kinetics (*Nagel and Wilson, 2011*) and may induce differing amounts of adaptation (*Cao et al., 2016*). Differences in temporal processing of odors across species could also therefore reflect differences in the kinetics of individual odor-receptor interactions. Experiments expressing moth receptors in fly neurons, or comparing the history-dependence of flight vs walking reactions in the same species, may help resolve these differences. Rapid odor fluctuations have also been observed to impair upwind progress in some moth species (*Riffell et al., 2014*).

## Modeling olfactory search behavior

To relate elementary sensory-motor transformations to behavior in complex odor environments, we developed a simple model of olfactory navigation. In our model, different forms of sensation, such as flow sensing (wind), temporal sensing (offset response) and spatial sensing (comparisons across the antennae) each produce distinct changes in forward and in angular velocity. The contributions of each form of sensing are summed to generate total turning behavior. Our model differs from previous models of turbulent navigation (*Pyk et al., 2006*; *Balkovsky and Shraiman, 2002*; *van Breugel and Dickinson, 2014*) in that it does not specify any distinct behavioral states such as 'upwind orientation' or 'casting.' This is consistent with the observation that intermediate behavior, in which a positive upwind velocity overlaps with an increase in angular velocity, can be observed during decreasing odor ramps. Our model also differs from those requiring the animal to derive and maintain an estimate of the source position (*Vergassola et al., 2007*; *Masson, 2013*). The only 'memory' required by our model is a slow adaptation and an offset response with a long integration time. Slow adaptation has been observed in the responses of olfactory receptor neurons and projection neurons (*Kaissling et al., 1987*; *Nagel and Wilson, 2011*; *Nagel et al., 2015*; *Cafaro, 2016*; *Gorur-Shandilya et al., 2017*), while offset responses with long integration times have been observed in antennal lobe interneurons (*Nagel and Wilson, 2016*). Thus, both these types of history-dependence have been experimentally demonstrated.

To validate our model, we showed that it can reproduce several features of experimentally observed fly behavior. First, the model can produce the upwind run during odor and the local search at offset that we observe in response to odor pulses in our miniature wind-tunnels. Second, it can

produce straighter trajectories during odor and local search in the absence of wind information. Third, variation in the scale of the ON and OFF functions can generate the type of variability we observe in behavior across flies. Finally, the model produces a distribution of behaviors (source finding, intermediate search, and downwind orientation) similar to that of real flies when tested in a turbulent odor plume. Despite these similarities, there are aspects of fly behavior that our model does not capture. For example, we were unable to precisely match the distribution of angular velocities observed in our data and still produce realistic trajectories. This suggests that there is additional temporal structure in real fly behavior that our model lacks. There are also discrepancies between our model predictions and the timing of responses near odor onset (particularly in the frequency sweep responses) that might reflect the simplicity of the filter model used, or might reflect real variability in the latency of flies to respond to odor. Nevertheless, our model provides a relatively straightforward way to understand the relationship between temporal filtering of odors, sensory-motor coupling, and behavior in various odor environments. It should thus facilitate studies relating changes in neural processing to olfactory behavior.

A question left open by our model is the role of spatial sensing (bilateral comparisons) in guiding navigation. We found that if the gain was set high enough, this form of sampling could significantly improve the model's performance (unrealistic gain values, of 1500 deg/s, could produce performance rates of over 95% success). This result is surprising, as previous studies have concluded that nearby samples taken in turbulent plume do not contain usable information (*Borst and Heisenberg, 1982*). However, recent studies have suggested that plumes may contain more usable spatial information than previously thought (*Boie et al., 2018*), particularly when the plume forms near a solid boundary (*Gire et al., 2016*). Using average gain values estimated from studies in tethered flies on a trackball (*Borst and Heisenberg, 1982*; *Gaudry et al., 2013*) we found that bilateral sampling contributed fairly little to performance, because the concentration differences across the antennae were typically quite small. In previous studies, bilateral sampling has been investigated largely using long-lasting odor stimuli of fixed concentration. It would be interesting in the future to ask whether flies can respond more strongly to small concentration differences when they are embedded in a fluctuating environment like the one measured here.

## Materials and methods

**Key resources table**

| Reagent type or resource | Designation | Source or reference | Identifiers | Additional information |
|---|---|---|---|---|
| Gene (*Drosophila melanogaster*) | norpA$^{36}$ | NA | FLYB:FBgn0262738 | — |
| Genetic reagent (*Drosophila melanogaster*) | w$^{1118}$ norpA$^{36}$ | This paper | FLYB:FBal0013129 | Progenitor = norpA$^{36}$ obtained from C. Desplan; backcrossed 7 generations to Bloomington stock 5905 = w$^{1118}$ |

### Fly strains

We used genetically blind *norpA$^{36}$* mutants, (*Ostroy and Pak, 1974*; *Pearn et al., 1996*) to avoid visual contributions to behavior. The *norpA$^{36}$* allele was backcrossed for seven generations to an isogenic *w$^{1118}$* stock (Bloomington 5905, also known as *iso31* as described in (*Ryder et al., 2004*) that exhibits robust walking behavior (*Stavropoulos and Young, 2011*), using PCR to follow the allele through backcrossing. *norpA$^{36}$* males were crossed to *w$^{1118}$* virgins and virgin female *norpA$^{36}$/+* progeny were backcrossed to *w$^{1118}$* males. In each subsequent generation, 15 to 20 virgin females were backcrossed singly to *w$^{1118}$* males and genomic DNA was extracted from each female after several days of mating. PCR amplification was performed with primers flanking the *norpA$^{36}$* deletion (*oNS659* AAACCGGATTTCATGCGTCG and *oNS660* TGTCCGAGGGCAATCCAAAC; 95C 2 min, 30x(95C 20 s, 60C 10 s, 72C 15 s, 72C 10 min) to identify heterozygous *norpA$^{36}$/+* mothers giving rise to wild-type (172 bp) and mutant (144 bp) products. After seven generations of backcrossing, single males were crossed to an isogenic *FM7* stock to generate homozygous stocks, and those bearing *norpA$^{36}$* were identified with PCR. Both *w$^{1118}$ norpA$^{36}$* and *w$^+$ norpA$^{36}$* stocks were

generated during backcrossing. We used only $w^{1118}$ $norpA^{36}$ flies for behavior. For this reason, we used $w^{1118}$ flies as 'sighted' controls, although the $w^{1118}$ allele does affect vision as well.

All flies were collected at least 1 day post-eclosion. After collection, flies were housed in custom-made cardboard boxes at room temperature (21.5-23.5C), with a light cycle of 12 hr, for at least 3 days prior to experiments to allow habituation. Different boxes were shifted by two hours relative to the others to allow us to perform several experiments with the same conditions in the same day. At the time of the experiments, flies were 5 to 14 days old (average age was 7.1±1.8 days). Prior to the experiments, flies were starved for 5 hr in an empty transparent polystyrene vial with a small piece of paper soaked in distilled water to humidify the air. Experiments were performed between 2–4 hr after lights on (ZT 2-ZT 4).

## Behavioral apparatus

Our behavioral apparatus (*Alvarez-Salvado and Nagel, 2018*) was modified from the design of *Bell and Wilson (2016)* and was designed to allow us to monitor the position and orientation of flies walking freely in two dimensions while tightly controlling the odor and wind stimuli they experienced. The behavioral arena was composed of several layers of laser-cut plastic, all 30 by 30 cm in size with varying thicknesses (detailed below), in which different shapes were cut to create an internal air circuit and four individual behavioral chambers that measured 14 by 4 by 0.17 cm each. The arena was designed using Adobe Illustrator (design: Adobe Systems, San Jose, CA; plastics: Pololu Corp, Las Vegas, NV and McMaster, Robbinsville, NJ; laser cutting: Pololu). The internal layers —in which the individual chambers were cut— were made of 0.5 mm-thick PETG (McMaster reference: 9513K123), 0.8 mm delrin (McMaster: 8575K131), and 0.4 mm fluorosilicone rubber (McMaster: 2183T11). Additionally, the arenas had a floor and ceiling layers made of 4.5 mm clear acrylic (Pololu).

The ceiling was held in place with seven set screws; combined with the fluorosilicone rubber layer this ensured that air did not escape from the chambers and produced more uniform odor concentrations throughout the arena. Each behavioral chamber had a separate air inlet through which charcoal-filtered air was supplied, and an outlet at the opposite end. A series of baffles in the PETG layer, as well as the short vertical extent of the chambers (1.7 mm) ensured laminar flow of air through our chambers (calculated Reynolds number 11.5). Total airflow through the arena, as measured by anemometer, was 11.9cm/s.

The arena was placed in an imaging chamber constructed from a breadboard (Thorlabs) and 80/20 posts (McMaster: 47065T101) held in place with brackets (McMaster: 47065T236). Illumination was provided by an LED panel composed of an aluminum sheet (McMaster: 88835K15 ) covered with infrared (IR) LED strips (Environmental lights, irrf850-5050-60-reel). A diffuser (Acrylite: WD008) was placed between the LED panel and the arena to provide uniform lighting. Flies were imaged from below the arena using a monochrome USB 3.0 camera (Basler: acA1920-155um) and a 12 mm 2/3'' lens (Computar: M1214-MP2). An IR filter (Eplastics: ACRY31430.125PM) was placed between the camera and the arena. LEDs were controlled using buckblock drivers (Digikey). An Arduino microprocessor (teensy 2.0, PJRC) was used to strobe the IR LEDs at 50 Hz and to synchronize them with each camera frame.

Imaging and stimulus delivery were controlled by custom software written in Labview (National Instruments, Austin, TX). Timing of odor was controlled by a National Instruments board (PCIe-6321). Flies were tracked by comparing the camera image at each time point to a background image taken prior to the experiment. Background-subtracted images were thresholded and binarized; a region of interest per chamber was then taken for further processing. Particle filtering functions were applied to each region of interest to remove particles less than 3 pixels (0.4 mm) long or greater than 50 pixels (6.8 mm) long. A particle analysis function was used to identify the fly in each chamber and to compute its center of mass and orientation.

Since the particle analysis function could only determine the fly's orientation up to 180 (i.e. it cannot distinguish the front and back of the fly), we used a second algorithm to uniquely determine the animal's orientation. Each background-subtracted image was passed through a second thresholding operation with a lower threshold intended to include the translucent wings. The center of mass of this larger particle was compared to the center of mass of the smaller wingless particle to determine the orientation of the fly in 360. Orientation measurements were strongly correlated with movement direction, but provided a smoother readout of heading direction when its velocity was low. Position

(X and Y coordinates) and orientation were computed in real time during data collection and saved to disk.

## Stimulus delivery

Wind and odor stimuli were delivered through inlets at the upwind end of the arena. Each arena was supplied with a main air line that provided charcoal-filtered wind. Wind flow rate was set to 1 L/min by a flowmeter (Cole-Parmer, Vernon Hills, IL). This line could be shut off by a three-way solenoid valve (Cole Parmer, SK-01540-11) in order to measure behavior in the absence of wind (*Figure 2*). To measure air flow, we used an anemometer (miniCTA 54T30, Dantec Dynamics, Skovlunde, Denmark), inserting the probe into the chambers through holes on a special ceiling made for this purpose. The anemometer was calibrated by measuring the outlet of a single 25 mm diameter tube (filled with straws to laminarize flow) connected directly to a flow meter. The measured air velocity was 11.9 cm/s.

Odor was delivered via rapidly switching three-way solenoid valves (LHDA1233115H, The Lee Company, Westbrook, CT) located just below the arena, that directed odorized air either to the chambers or to a vacuum. Each chamber had its own valve, and odor was injected just downstream of the main air inlet, 1.7 cm upstream of the baffle region of the chamber. Charcoal-filtered air was odorized by passing it through a scintillation vial filled with 20 ml of odorant solution (apple cider vinegar or ethanol), then directed through a manifold (McMaster: 4839K721) to each of the four valves. Importantly, the vials containing the odor solution were almost full, creating a relatively small head space where odor could readily accumulate. Odorized air flow rate was set to 0.4 l/min using flowmeters. During non-odor periods, odorized air was directed into a vacuum manifold and away from the apparatus. Flow rates in the arena and vacuum manifold were matched to eliminate transients in odor concentration during switching. An equal volume of 'balancing' air was injected into the arena during these periods to maintain a constant air flow rate throughout the experiment. Balancing air was humidified by passing over an identical scintillation vial filled with water and was delivered by an identical three-way valve. Odor and balancing valves fed into a small t-connector, that was suspended from the arena using $\approx$ 1 cm of tygon tubing (0.8 mm inner diameter, E-3603). This design, in which odor flowed continuously and was switched close to the arena, produced rapid odor dynamics with few concentration artifacts, but also a small mechanical stimulus when the valve was switched. This odor delivery system was using for experiments in *Figures 1* and *2*, and for intensity experiments in *Figure 3A–B*.

To produce analog odor stimuli including ramps, frequency sweeps, and the plume walk stimulus, we added two-way proportional valves (EVP series, EV-P-05-0905; Clippard Instrument Laboratory, Inc., Cincinnati, Ohio) 20 cm upstream of the odor and balancing scintillation vials. Proportional valves were driven indepentendly by valve drivers (EVPD-2; Clippard) and were calibrated so that their maximal opening would produce the same flow rate as in experiments using three-way valves. (three-way valves were held open during experiments with analog stimuli.) Proportional valves produce increasing airflow with applied current; however, they exhibit both nonlinearity and histerisis, in which the effect of a driving current depends on the past and current state of the valve. To generate our desired stimulus waveforms, we first provided an ascending and descending ramp stimulus to the valves and measured the subsequent odor waveform in the behavioral chambers using a PID (see below). We used the results of that measurement as a lookup table to create a driving current command that produced the desired odor waveforms. Lookup tables for odor and balancing valves were measured separately. We used PID measurements at several locations in the arena to verify that the delivered odor waveform matched our desired odor waveform. We used an anemometer (see below) to verify that the total flow rate during the stimulus (in which odor and balancer valves were run together) did not vary by more than 1%.

To measure odor concentration in our arenas we used a photo-ionization detector (miniPID, Aurora Systems, Aurora, Canada) inserted into the arena, again using a special ceiling. All calibration measurements were made using 10% ethanol, which provided higher signal to noise than ACV. Measurements at the top of the arenas revealed an average rise time of $\approx$ 180 ms and a fall time of $\approx$ 220 ms for square pulses delivered using three-way valves. The latency of the measured odor onset from nominal odor onset increased linearly with distance from the odor source (up to 900-1000 ms at the downwind end of the arena), consistent with our measurement of air velocity (*Figure 1—figure supplement 1*). For frequency sweep stimuli, we observed some widening of peaks

with distance down the arena, consistent with the effects of diffusion (*Figure 1—figure supplement 1*). Diffusion thus set the upper limit on the frequency of stimuli that we could reliably deliver within our arena (about 1Hz). Presenting higher frequency stimuli would require higher wind speeds, but we found that higher wind speeds caused flies to stop moving, as previously observed (*Yorozu et al., 2009*).

## Experimental protocol

Each experiment lasted approximately 2 hr, during which flies performed an average of 86.7±7.7 trials. (Some trials were discarded due to tracking problems, as described below, and not all experiments lasted exactly the same amount of time). Each trial lasted 70 s, and was followed by a gap of ≈6 seconds while the computer switched to the next trial. There were three to four types of trials that were randomly interleaved during the experiment. One of those types was always a blank trial, in which flies only experienced clean air flow. The other types corresponded to different types of odor stimuli, that were dependent on the experiment: namely, square odor pulses for experiments in *Figures 1*, *2* and *3A–B*; odor ramps in *Figure 3C–D*; frequency sweeps and plume data in *Figure 3E–J*. To ensure repeatability, data for all experiments was collected over several different days (5 to 9, often non-consecutive). For *Figure 1*, we used data from experiments performed over a period of 7 months.

For experiments in *Figure 2*, we rendered flies 'wind-blind' by anesthetizing them on a cold plate and cutting their aristae and stabilizing their antennae. We cut the aristae by clipping them with fine forceps at the lowest possible level without touching the antennae. Then, we put a very small drop of ultra-violet (UV) glue on the anterior side of the antennae, falling between the second and third segments, as well as touching the rest of the clipped aristae. We then used a pen-sized ultra-violet light to cure the glue, and made sure it was solid before putting the flies back to their home vials to recover for 24 hr. The whole procedure took approximately 5 min, and never longer than 10. We did this procedure in a pair of flies at a time, stabilizing the antenna of one and using the other as sham-treated (it was placed on the cold plate and under the UV light exactly like the treated fly was).

For experiments in *Figure 6*, approximately 48 hr before the experiment, we applied a drop of UV glue connecting both wings of the fly or to each wing hinge. This prevented flies from flying while allowing us to still use the wings to detect heading.

## Analysis of behavioral data

All analyses were performed in Matlab (Mathworks, Natick, MA) (*Alvarez-Salvado and Nagel, 2018*; copy archived at https://github.com/elifesciences-publications/AlvarezSalvado_ElementaryTransformations). X and Y coordinates and orientation information were extracted from the data files, and any trials with tracking errors (i.e. flies' position was missed at some point) were discarded (this occurred rarely). In some trials, we observed orientation errors in the form of sudden changes of approximately 180. In these cases, orientation was corrected by calculating the heading of the flies using X and Y coordinates, and filling in the gaps in orientation using the orientation that best correlated with that information, producing coherent and continuous orientation vectors. Coordinates and orientations were low-pass filtered at 2.5 Hz using a two-pole Butterworth filter to remove tracking noise that was produced especially when flies were not moving. X and Y coordinates were then converted to mm, and trials in which flies moved less than a total of 25 mm were discarded. Distance moved was calculated as the length of the hypotenuse between two subsequent pairs of coordinates.

We next calculated a series of gait parameters from each trial's data. Ground speed was obtained by dividing the distance moved by the time interval of each frame (20 ms). Upwind velocity was calculated using the derivative of the filtered Y coordinates divided by the time interval of 20 ms. Angular velocity was calculated as the absolute value of the derivative of the filtered unwrapped orientation (i.e. orientation with phases corrected to be continuous beyond 0° or 360°) divided by the time interval of 20 ms. For all gait parameters shown (ground speed, upwind velocity, angular velocity), we excluded data points in which ground speed was less than 1 mm. This was necessary because flies spend a large amount of time standing still. Distributions of gait parameters are therefore highly non-Gaussian, with large peaks at 0 (*Figure 3—figure supplement 1A*), and parameter means are highly influenced by the number of zeros. In addition, the probability of moving (obtained

by binarizing the ground speed with a threshold of 1 mm/s) changes dramatically in response to odor, and remains high for tens of second after odor offset (*Figure 3—figure supplement 1B*). Exclusion of the large number of zeros from average gait parameters produced more reliable estimates of these parameters. Curvature was calculated by dividing angular velocity by ground speed (excluding any points where ground speed was less than 1 mm/s). Turn probability was calculated binarizing curvature with a threshold of 20 deg/mm.

Because it takes a little over a second for the odor waveform to advect down the arena, the exact time of odor encounter and loss depends on the position of the fly within the arena. This advection delay has a strong effect on our estimates of gait parameter dynamics, particularly for fluctuating sinusoidal stimuli. We therefore developed a warping procedure to align behavioral responses to the actual time at which each fly encountered the odor on each trial. To implement this procedure, we first recorded the PID response to each stimulus at three different points along the arena (*Figure 1—figure supplement 1*). We then calculated the delay for the odor to reach the position of the fly for each time frame during the odor stimulus, and shifted all the data points back by this amount. The periods before and after the odor stimulation are also shifted according to the initial position of flies in the odor period. This method can skip a data point when the fly moves upwind or can repeat a data point when the fly moves downwind, but the majority of the data are conserved and the resulting waveforms resemble very much the initial ones. After warping, all trials from all flies can be equally compared to a standard PID measurement done at the top of the arenas (i.e. the odor source). Warping was applied to all data shown in *Figures 1–3*. Note that in experiments using three-way valves (*Figure 1*), the click of the valve produced a brief freezing responses that was visible as a dip in ground speed. However, because of the warping, the time of the valve click is distributed across flies, as their ground speeds have been aligned to the time of odor encounter rather than the time of valve opening. This results in a smeared dip in the ground speed trace near the beginning and end of the odor stimulus.

For experiments using frequency sweeps and plume walks, we additionally excluded data obtained after the fly reached the top end of the chamber, as well as data from within 3 mm of the side walls. These exclusions were made to minimize the effect of arena geometry on gait parameter estimates, and to exclude regions where boundary layer effects would cause the odor waveform to advect more slowly. To calculate the data shown in the insets of *Figure 3E–H*, and in *Figure 3—figure supplement 1F*, we used a jackknife procedure to resample the responses of flies to frequency sweep stimuli. We made 10 estimates of the mean, excluding 34 trials from each estimate. To estimate the modulation of upwind velocity and ground speed in response to each cycle of the stimuli, we took the times between minima of the stimulus waveform as the limits for each cycle of the ascending frequency sweep; for the descending frequency sweep we used the intervals between maxima of the odor waveform. Within those limits, we calculated the minimum-to-maximum amplitude for each of the 10 different mean responses. The results shown in the figures are the mean of these estimates as a function of frequency of the corresponding stimulus cycles. The frequency of the cycles was estimated as 1 over the duration of the cycle. Error bars in the figure insets represent the standard error (*SE*) across estimates, calculated as

$$SE = \frac{\sqrt{\frac{n-1}{n}\sum_{i=1}^{n}(\overline{x}_i - \overline{x})^2}}{\sqrt{n}} \tag{5}$$

where $\overline{x}_i$ is each of the peak-to-peak estimates excluding one fly, $\overline{x}$ the estimate including all flies, and $n$ the number of data subsets used (10).

## Statistical analysis

In *Figure 1G*, *Figure 2B* and *Figure 5G*, we compared the mean values of different motor parameters from the same fly in three different periods of time in the trials, namely: 'before odor' from $-30$ to 0 s before the odor, 'during odor' from 2 to 3 seconds during the odor, and 'after odor' from 1 to 3 s after odor offset. We performed a Wilcoxon signed rank paired test for each of those comparisons and corrected the threshold for statistical significance *alpha* using the Bonferroni method. All significant comparisons were marked with asterisks in the figures, and the exact p values obtained are presented in the following tables.

| Comparison | Upwind velocity | Ground speed | Angular velocity | Curvature | Turn probability |
|---|---|---|---|---|---|
| Before–during odor | $2.0 \cdot 10^{-12}$ | $3.9 \cdot 10^{-9}$ | $1.7 \cdot 10^{-3}$ | $4.9 \cdot 10^{-5}$ | $2.3 \cdot 10^{-3}$ |
| Before–after odor | $6.3 \cdot 10^{-2}$ | $7.7 \cdot 10^{-6}$ | $1.2 \cdot 10^{-11}$ | $5.5 \cdot 10^{-10}$ | $7.3 \cdot 10^{-14}$ |
| During–after odor | $1.4 \cdot 10^{-12}$ | $1.5 \cdot 10^{-10}$ | $9.5 \cdot 10^{-11}$ | $7.1 \cdot 10^{-10}$ | $4.8 \cdot 10^{-12}$ |

p values for comparisons made in *Figure 1G*. The *alpha* value after correcting for multiple comparisons was 0.0167.

| Comparison | Upwind velocity | Ground speed | Curvature |
|---|---|---|---|
| Before–during odor | 0.27 | 0.016 | 0.34 |
| Before–after odor | 0.84 | 0.85 | 0.003 |
| During–after odor | 0.41 | 0.008 | 0.002 |

p values for comparisons made in *Figure 2B*. The *alpha* value after correcting for multiple comparisons was 0.0167.

| Comparison | Upwind velocity | Turn probability |
|---|---|---|
| Before–during odor | $1.3 \cdot 10^{-83}$ | $1.2 \cdot 10^{-55}$ |
| Before–after odor | $9.0 \cdot 10^{-46}$ | $1.3 \cdot 10^{-83}$ |
| During–after odor | $1.3 \cdot 10^{-83}$ | $1.3 \cdot 10^{-83}$ |

p values for comparisons made in *Figure 5G*. The *alpha* value after correcting for multiple comparisons was 0.0001.

| Comparison | Upwind velocity | Ground speed | Angular velocity | Curvature | Turn probability |
|---|---|---|---|---|---|
| Before–during odor | $9.4 \cdot 10^{-11}$ | $4.6 \cdot 10^{-7}$ | $5.6 \cdot 10^{-1}$ | $1.0 \cdot 10^{-1}$ | $3.2 \cdot 10^{-6}$ |
| Before–after odor | $3.7 \cdot 10^{-9}$ | $1.7 \cdot 10^{-5}$ | $5.2 \cdot 10^{-5}$ | $2.1 \cdot 10^{-6}$ | $8.1 \cdot 10^{-10}$ |
| During–after odor | $3.2 \cdot 10^{-9}$ | $5.2 \cdot 10^{-9}$ | $3.1 \cdot 10^{-3}$ | $4.9 \cdot 10^{-6}$ | $9.3 \cdot 10^{-5}$ |

p values for comparisons made in *Figure 1—figure supplement 3B*. The *alpha* value after correcting for multiple comparisons was 0.0167.

To estimate the Standard Error of the proportion of successful trials shown in Figure 6I and in Figure 7C, we used the formula

$$SE = \sqrt{\frac{p(1-p)}{n}} \tag{6}$$

where $p$ was the proportion of successful trials and $n$ the number of trials.

To test for statistical differences in *Figure 7C*, we calculated a $z$ statistic by normal approximation of the corresponding binomial distributions according to:

$$z = \frac{p_1 - p_2}{\sqrt{p(1-p)\left(\frac{1}{n_1} + \frac{1}{n_2}\right)}} \tag{7}$$

where $p_1$ and $p_2$ are the probabilities of success in the two distributions being compared, $p$ is the probability of both distributions combined, and $n_1$ and $n_2$ are the number of trials in the two distributions. We then estimated the p values by evaluating a normal cumulative distribution function of a standard normal distribution for the resulting $z$ values. This analysis yielded the following results:

| Comparison | z statistic | p value |
|---|---|---|
| $\kappa_7 = 0$ **VS** $\kappa_7 = 40deg/s$ | 1.03 | 0.30062 |
| $\kappa_7 = 0$ **VS** $\kappa_7 = 300deg/s$ | 3.50 | 0.00046 |
| $\kappa_7 = 0$ **VS** $\kappa_7 = 300deg/sSWAP$ | 4.12 | 0.00004 |

*z* statistics and p values for comparisons made in **Figure 7C**. The *alpha* level used was 0.05.

## Computational modeling

Our computational model was composed of two parts (**Alvarez-Salvado and Nagel, 2018**). In the first, we asked whether simple phenomenological models, comprised of a linear filtering step, and a nonlinear adaptive compression function, were capable of capturing the dynamics of upwind velocity and turn probability in response to a wide array of odor waveforms. We compared fits of four model versions to our behavioral data, and tested the resulting best fit model by predicting responses to the plume walk stimulus. These fits comprise the two temporal functions which we call ON and OFF.

In the second part, we asked whether a simple navigational model, based on the ON and OFF functions fit to the data and described in **Figure 5**, was capable of reproducing the types of trajectories we observed experimentally and of locating the source of a real odor plume. In addition, this model allowed us to test the contribution of each of its components to successful odor localization. In the model, we first compute two temporal functions of the odor stimulus, ON and OFF. These two signals are then used to modulate ongoing behavioral components (angular velocity and ground speed) which iteratively update the fly's position. The model can be run in open loop, as in our behavioral expeirments, by providing an odor input as a function of time, or in closed loop, where the odor concentration at any point in time depends on the fly's position in a real or virtual space. All computational modelings were performed in Matlab. Differential equations were simulated using the Euler method with a time step of 20 ms.

### Odor ON and OFF functions

The ON function was composed of an adaptive compression step and a linear filtering step (model ACF in **Figure 4**). Adaptation was driven by an adaptation factor A(t) that accumulated slowly in the presence of odor:

$$\tau_A \frac{dA}{dt} = odor(t) - A(t) \tag{8}$$

Compression was modeled using a Hill equation with a baseline $\kappa_d$ of 0.01 (expressed as a fraction of our highest odor concentration: 10% apple cider vinegar). This baseline value was taken from our fits of responses to square pulses of different concentration (**Figure 3**). Adaptation slowly increased the effective $\kappa_d$, reducing the sensitivity of behavior to odorant, and maintaining responses of about the same size over a wide concentration range:

$$C(t) = \frac{odor(t)}{odor(t) + \kappa_d + A(t)} \tag{9}$$

The filtering step was given by

$$\tau_{ON} \frac{dON}{dt} = C(t) - ON(t) \tag{10}$$

For the OFF model, adaptation and compression were modeled in the same way, but filtering was performed by applying two filters, one fast and one slow, and then taking the difference between the slow and the fast filter output, thresholded at 0:

$$\tau_{fast} \frac{dR1}{dt} = C(t) - R1(t) \tag{11}$$

$$\tau_{slow} \frac{dR2}{dt} = C(t) - R2(t) \tag{12}$$

$$OFF = max(0, R2 - R1) \tag{13}$$

Model parameters used in *Figure 4* are shown in *Table 1*. These same model parameters were used for all remaining simulations. We also considered three additional models. In the FAC model, the order of operations was inverted, so the odor was first filtered, then adaptively compressed. In the CF and FC models, we omitted the adaptation step, and again tried both orders of operation (compression first and filtering first):

$$C(t) = \frac{odor(t)}{odor(t) + \kappa_d} \tag{14}$$

We found that models lacking adaptation performed significantly worse for both ON and OFF. All fits were made using the function *nlintfit* in Matlab. Fit parameters and RMSE values are given in *Table 1*.

## Modulation of behavioral components

The temporal functions described above were used to modulate the ground speed of the fly $v$ and its heading $H$, from which the XY coordinates of the position of the fly at each point in time could be calculated.

The ground speed at each time step was give by:

$$v(t) = v_0 + \kappa_1 ON(t) - \kappa_2 OFF(t), \text{where } v \geq 0 \tag{15}$$

where $v_0$ is the baseline speed, set at 6 mm/s based in our behavioral data. $\kappa_1$ and $\kappa_2$ determine the influence of ON and OFF functions on the final speed.

The heading at each time step ($\Delta t$ of 20 ms) was computed by adding the instantaneous angular velocity to the current heading:

$$H(t + \Delta t) = H(t) + \Delta t \cdot \dot{\theta}(t) \tag{16}$$

The angular velocity at each time step $\dot{\theta}(t)$ is a linear sum of several components driven by different sensory stimuli: a random component, driven by odor dynamics, and two deterministic components, driven by wind:

$$\dot{\theta}(t) = \rho(t)G(0,\sigma)^2 + \kappa_5 ON(t)D_u(\psi) + \kappa_6 D_d(\psi) \tag{17}$$

The first term represents probabilistic turning whose rate is modulated by the dynamics of odor. $\rho(t)$ is a binary Poisson variable that generates a draw from a Gaussian distribution with mean 0 and standard deviation $\sigma$ when it is equal to 1. The value drawn from this distribution was squared to yield a distribution of angular velocities with higher kurtosis, as observed in the distribution of real flies' angular velocities. However, we did not attempt to directly match the distribution of angular velocities found in our data. (Indeed, we found that matching this distribution produced trajectories that were far too jagged, suggesting that one of the assumptions of the model, for example that angular velocities are independent of one another, or that angular velocity is independent of forward velocity, must be incorrect.) The rate of $\rho(t)$ is given by

$$P(t) = P_0 - \kappa_3 ON(t) + \kappa_4 OFF(t) \tag{18}$$

Thus, the rate of random turns has a baseline of $P_0$, decreases in the presence of odor (when ON (t) is positive) and increases after odor offset (when OFF(t) is positive).

The second and third terms represent deterministic turns driven by wind. To model these turns, we defined two sinusoidal functions —$D_u$ for upwind orientation and $D_d$ for downwind orientation— given by the equations:

$$D_u(\psi) = \sin(\psi) \tag{19}$$

$$D_d(\psi) = -\sin(\psi) \tag{20}$$

where $\psi$ is the direction of the wind relative to the fly. A negative value of $\psi$ indicates wind coming from the fly's left, and a positive value of $\dot{\theta}(t)$ indicates a turn to the left, so $D_u$ produces a turn to the left when wind is sensed on the left and vice-versa, leading to upwind orientation. The function $D_d$ produces a turn to the right when wind is sensed on the left resulting in downwind orientation. The downwind function $D_d$ is always on but has a small coefficient, resulting in a mild downwind bias when combined with baseline random turning driven by the first term $\rho(t)G(0,\sigma)$. The upwind function $D_u$ is gated by the ON function and has a larger coefficient. This means that in the presence of odor, this term comes to dominate turning, driving strong upwind orientation.

To estimate values for the coefficients $\kappa_1$ to $\kappa_6$ we ran simulations of the model using a 10 s odor pulse and adjusted parameters sequentially so that analysis of the model outputs matched as closely as possible the response of real flies shown in *Figure 5E*. We first adjusted $\kappa_1$ and $\kappa_2$ to match the ground speed. Next, we adjusted $\kappa_5$ and $\kappa_6$ to match the upwind velocity. Finally, we adjusted $\kappa_3$ and $\kappa_4$ to match turn probability. We matched the theoretical turn probability on the model (the rate governing the Poisson variable $\rho(t)$ rather than the turn probability extracted from the model trajectories.

To generate trajectories with this model, the X and Y coordinates were calculated from $v(t)$ and $H(t)$, according to:

$$X(t+\Delta t) = X(t) + \Delta t \cdot v(t) \cos\left(H(t)\right) \tag{21}$$

$$Y(t+\Delta t) = Y(t) + \Delta t \cdot v(t) \sin\left(H(t)\right) \tag{22}$$

Simulations in the turbulent plume were run at 15 Hz rather than 50 Hz to match the sample rate of the plume measurements. All rate constants (including turn probability per sample) were converted accordingly. *Video 2* shows an example of the model navigating a real odor plume.

To add bilateral sensing to our navigation model in *Figure 7*, we made two measurements of odor concentration at each point in time. $Odor_L$ was the odor concentration at the location of the fly in the plume movie, while $odor_R$ was the concentration one pixel (740 $\mu m$) to the right. This spacing is perhaps twice the distance between a fruit fly's antennae, but represents the closest sampling we could perform using our current imaging system. We then applied the adaptive compression given by *Equations 17 and 18* to each odor measurement separately:

$$C_L(t) = \frac{odor_L(t)}{odor_L(t) + \kappa_d + A(t)} \tag{23}$$

$$C_R(t) = \frac{odor_R(t)}{odor_R(t) + \kappa_d + A(t)} \tag{24}$$

The bilateral contribution to angular velocity was computed as the difference between the two compressed odor signals, multiplied by a coefficient $\kappa_7$ that determines how strongly the fly turns when it detects a concentration difference. We estimated $\kappa_7$ from the literature (*Borst and Heisenberg, 1982*; *Cardé and Willis, 2008*) by examining the turn rates produced when a maximal concentration difference $(C_L - C_R = 1)$ was applied across the antennae. The bilateral contribution was added as a fourth component to the equation governing total angular velocity (*Equation 17*):

$$\dot{\theta}(t) = \rho(t)G + \kappa_5 ON(t)D_u(\varphi) + \kappa_6 D_d(\varphi) + \kappa_7(C_L - C_R) \tag{25}$$

## Turbulent wind tunnel construction

We generated a turbulent odor plume in a low-speed bench-top wind tunnel with a flow-through design. Two wind tunnels were built, one in Colorado (for plume measurements) and one in New York (for behavioral measurements). In the Colorado wind tunnel, air entered the tunnel through a bell-shaped contraction (4:1 ratio) and passed through a turbulence grid (6.4 mm diameter rods with a 25.5 mm mesh spacing) prior to the test section. The test section was 30 cm wide, 30 cm tall, and extended 100 cm in the direction of the flow. Air exited the test section through a 15 cm long honeycomb section used to isolate the test section from a fan located in the downstream contraction. The fan generated a mean flow of air through the tunnel at 10 cm/s. Acetone was released isokinetically into the center of the test section through a 0.9 cm diameter tube aligned with the flow. The

tube opening was located 10 cm downstream of the turbulence grid and 6 mm above a false floor spanning the length and width of the test section. The New York tunnel was designed similarly, except that test section measured 38 cm by 38 cm by 92 cm, the honeycomb was 5 cm long, and odor was released from a 1cm diameter tube at floor level. Air flow was 10cm/s and odor release was isokinetic as in the Colorado wind tunnel. The New York tunnel was fitted with an aluminum IR light panel (Environmental lights, irrf850-5050-60-reel) 2.5 cm below a diffuser (Acrylite: WD008) and a 1 cm thick acrylic layer that acted as the arena floor. A channel 1 cm wide and 0.4 cm deep was milled into this arena and filled with water to constrain flies to walk within the imaging area, 31 cm wide and 87 cm long. Two cameras (Point Grey: 2.3MP Mono Grasshopper3 USB 3.0) with 12 mm 2/3' lenses (Computar: M1214-MP2) were suspended approximately 45 cm above the arena to image fly movement. Tracking code was written in Labview and used the same algorithms as described above to extract position and heading at 50Hz.

### Plume measurements in air

To measure plume structure and dynamics in air, we used a planar laser-induced fluorescence (PLIF) system (*Lozano et al., 1992*) to image a plume of acetone vapor (*Connor et al., 2018*). A UV laser light sheet entered the test section of the tunnel through a slit along the length of the test section to excite acetone vapor. A camera imaged the resulting acetone fluorescence in the test section through a glass window. The imaging area covered up to 30 cm downwind from the odor source and up to 8 cm to both sides. The plume was imaged in the 1-mm-thick laser sheet centered on the tube 6 mm above the bed. A total of 4 min were recorded. Images were then post-processed into calibrated matrices of normalized concentrations.

We produced acetone vapor by bubbling an air and helium gas mixture through flasks partially filled with liquid acetone. To reduce fluctuations in concentration, a water bath maintained flask temperature at 19 deg C which was approximately 2 degrees cooler than ambient air temperature to prevent condensation. To account for the density of acetone, we blended air (59% v/v) and helium (41% v/v) for the carrier gas. Assuming 95% saturation after contact with the liquid acetone, the gas mixture was approximately 25% acetone by volume and neutrally buoyant.

An Nd:YAG pulsed laser emitted light at a wavelength of 266 nm and a frequency of 15 Hz to illuminate the acetone plume. After excitation at that wavelength, acetone vapor fluoresces with an intensity proportional to its concentration. A high-quantum efficiency sCMOS camera imaged the acetone plume fluorescence at 15 Hz. To enhance signal and minimize noise, we collected data in a dark environment, used a lens with high light-gathering capabilities (f/0.95), and binned the pixels from 2048x2048 native resolution to 512x512 resolution (0.74 mm/pixel).

Images were post-processed using an algorithm adapted from (*Crimaldi, 2008*) to correct for variations in laser sheet intensity, lens vignette, and pixel-to-pixel gain variation. The correction used a spatial map of the image system response by imaging the test section while it was filled with a constant and uniform distribution of acetone. Signal intensities were normalized by the intensity at the tube exit such that concentrations have average values between 0 and 1.

The 'plume walk' stimulus was generated by taking the time course of odor concentration along a linear trajectory going upwind through a plume movie at 6 mm/s (the average ground speed of our flies), starting 8.9 cm laterally from the midline and 30 cm downwind from the source.

## Acknowledgements

The authors thank D Schoppik, B Ermentrout, J Verhagen, N Urban, L Jacobs, and members of the Nagel and Schoppik labs for helpful comments and input during this project. J Tuthill and the Tuthill lab read an earlier version of the manuscript. J Bell and R Wilson shared the design of their behavioral apparatus (on which ours is based) prior to publication. This work was supported by NSF IOS-1555933, NIDCD R00DC012065, NIMH R01MH109690, and fellowships from the Klingenstein-Simons, Sloan, and McKnight Foundations to KIN, by NSF PHY-155586 to JPC and JDV, and by the Mathers, Whitehall, Alfred P Sloan, and Leon Levy Foundations, a NARSAD Young Investigator Award from the Brain and Behavior Research Foundation, an NYU Whitehead Fellowship, the J Christian Gillin, MD Research Award from the Sleep Research Society Foundation, and the Irma T Hirschl / Weill-Caulier Career Scientist Award to NS.

## Additional information

### Funding

| Funder | Grant reference number | Author |
|---|---|---|
| Whitehall Foundation | 2013-05-78 | Nicholas Stavropoulos |
| Leon Levy Foundation | Leon Levy Fellowship | Nicholas Stavropoulos |
| Brain and Behavior Research Foundation | NARSAD Young Investigator Award | Nicholas Stavropoulos |
| G Harold and Leila Y. Mathers Foundation | Mathers Award | Nicholas Stavropoulos |
| New York University | Whitehead Fellowship | Nicholas Stavropoulos |
| Sleep Research Society | J. Christian Gillin, M.D. Research Award | Nicholas Stavropoulos |
| Irma T. Hirschl Trust | Career Scientist Award | Nicholas Stavropoulos |
| Alfred P. Sloan Foundation | Sloan Foundation Fellowship | Nicholas Stavropoulos |
| National Science Foundation | PHY-1555933 | Jonathan D Victor |
| National Science Foundation | PHY-155586 | John P Crimaldi |
| National Science Foundation | IOS-1555933 | Katherine I Nagel |
| National Institutes of Health | R00DC012065, R01MH109690 | Katherine I Nagel |
| McKnight Endowment Fund for Neuroscience | McKnight Scholar Award | Katherine I Nagel |
| Alfred P. Sloan Foundation | Sloan Foundation Fellowship | Katherine I Nagel |
| Esther A. and Joseph Klingenstein Fund | Klingenstein-Simons Fellowship | Katherine I Nagel |

The funders had no role in study design, data collection and interpretation, or the decision to submit the work for publication.

### Author contributions

Efrén Álvarez-Salvado, Conceptualization, Data curation, Software, Formal analysis, Validation, Investigation, Visualization, Methodology, Writing—original draft, Writing—review and editing; Angela M Licata, Software, Formal analysis, Investigation, Methodology, Writing—review and editing; Erin G Connor, Formal analysis, Investigation, Methodology, Writing—review and editing; Margaret K McHugh, Formal analysis, Investigation, Methodology; Benjamin MN King, Resources; Nicholas Stavropoulos, Resources, Funding acquisition, Writing—review and editing; Jonathan D Victor, Conceptualization, Formal analysis, Supervision, Funding acquisition, Methodology, Project administration, Writing—review and editing; John P Crimaldi, Katherine I Nagel, Conceptualization, Software, Formal analysis, Supervision, Funding acquisition, Methodology, Writing—original draft, Project administration, Writing—review and editing

### Author ORCIDs

Nicholas Stavropoulos http://orcid.org/0000-0001-5915-2760
Jonathan D Victor https://orcid.org/0000-0002-9293-0111
Katherine I Nagel http://orcid.org/0000-0002-6701-3901

### Decision letter and Author response

Decision letter https://doi.org/10.7554/eLife.37815.024
Author response https://doi.org/10.7554/eLife.37815.025

## Additional files

### Supplementary files
• Transparent reporting form
DOI: https://doi.org/10.7554/eLife.37815.018

### Data availability
Code and files for behavioral paradigm, behavioral analysis, and modeling are available on GitHub at nagellab/AlvarezSalvado_ElementaryTransformations (copy archived at https://github.com/elifes-ciences-publications/AlvarezSalvado_ElementaryTransformations). Behavior and plume data are available at https://dx.doi.org/10.5061/dryad.g27mq71.

The following dataset was generated:

| Author(s) | Year | Dataset title | Dataset URL | Database, license, and accessibility information |
|---|---|---|---|---|
| Álvarez-Salvado E, Licata A, Connor E, McHugh M, King B, Stavropoulos N, Victor J, Crimaldi J, Nagel K | 2018 | Data from: Elementary sensory-motor transformations underlying olfactory navigation in walking fruit flies | https://dx.doi.org/10.5061/dryad.g27mq71 | Available at Dryad Digital Repository under a CC0 Public Domain Dedication |

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
