## [Decision Letter]

Thank you for submitting your article "Elementary sensory-motor transformations underlying olfactory navigation in walking fruit flies" for consideration by *eLife*. Your article has been reviewed by three peer reviewers, including Ronald L Calabrese as the Reviewing Editor and Reviewer #1, and the evaluation has been overseen by Eve Marder as the Senior Editor.

The reviewers have discussed the reviews with one another and the Reviewing Editor has drafted this decision to help you prepare a revised submission.

Summary:

In this manuscript, the authors present a creative analysis of fly walking navigation in response to an attractive odor (apple cider vinegar, ACV) in a mini wind tunnel that creates a boundary layer odor plume in which the fly is constrained to walk. Flies were placed in the rectangular arenas, where they were exposed to a constant flow of filtered, humidified air, defining the wind direction. Into this airflow pulses of odor were injected with rapid onset and offset kinetics, producing a front of odor that was transported down the arena at 11.9 cm/s. The odor dynamics were monitored by a photo-ionization detector (PID). They also constructed and tested flies in larger wind tunnels capable of delivering a turbulent boundary layer odor plume after the mini wind tunnel results. They also developed a simple computational model of how odor dynamics and wind direction influence changes in forward and angular velocity to simulate results in both types of wind tunnels. All flies tested were genetically blinded. Behavioral results with odor pulses show that flies have two basic responses with odor onset flies orient upwind and increase their ground speed (on response), and with odor offset, they reduce their ground speed and increase their rate of turning (off response). By mechanically blocking antennal wind sensation, they show that antennal mechanosensation is required for the directional components of these behaviors – on response, while odor offset is sufficient to induce changes in ground speed and turning – off response. Using time-varying odor stimuli, they show that the responses, on and off, low pass filter the odor dynamics and fitting their model they estimate 0.72 s and 4.84 s time constants respectively and that both responses show adaptation to odor with a time constant around 10 s. With the model they show that simulated flies respond similarly to the mean of all flies and that by simply changing the gains of the on and off responses they can capture individual variation about this mean. They then test files in the turbulent plume of the larger wind tunnel and find that their model can likewise capture their behavior including failures to find the odor source. They conclude that integration over time may be a useful computational strategy for navigating in a boundary layer plume, allowing flies to head upwind more continuously in the face of odor fluctuations, and to generate re-orientations clustered at the plume edges. The impact of the paper is in reducing walking navigation in a turbulent boundary layer to two basic stimulus driven responses one bi-modal (wind and odor on response) and one unimodal (odor off response) and providing a simple model to show how these responses interact in real odor plumes. They recognize that their analyses/conclusions are a first order approximation and that other variables/responses can be considered in future but nevertheless, the work is a major step forward in walking odor navigation and should be of wide interest in the behavioral neuroscience community.

Essential revisions:

1) As stated in the shortened review of reviewer 2, the authors should consider experiments in which one antenna is blocked; mechanoreceptors and potentially also olfactory receptors. These experiments will address the contribution of bilateral antennal sampling, which is known to be important in flies and other insects.

2) There are concerns that using only blind flies may influence the navigation strategy observed. Any data from sighted flies that could be included should be, or the potential implications of blindness should be more extensively discussed.

3) As stated in the shortened review of reviewer 3, the model should be better clarified and the parameters better rationalized.

4) The Discussion should more extensively discuss the type of navigation the flies are doing here better in the context of flight navigation.

Reviewer #2:

- In past work, several labs have reported the contribution of bilateral sampling (stereo-olfaction), from the pioneering work of Axel Borst in 1983 to the recent optogenetic-stimulation experiments conducted by Gaudry and Wilson. The fact that this aspect of the navigation mechanism is not addressed in the present manuscript represents a weakness. It seems that the author could have tested relatively easily the behavior of unilaterally wind-blind flies. They could then ask whether up-wind surges are possible with a single functional antenna, whether unilaterally wind-blind flies display a turn bias, etc. Likewise, unilateral olfactory impairment could have been produced through mechanical impairments (Duistermars and Frye, 2009). Combining these results in the model would nicely complete the present analysis. Although this weakness is minor, the authors are encouraged to address it.

- The use of blind (*norpA*) mutants in the olfactory navigation experiments is sensible. I was nonetheless wondering whether the authors have any evidence that the orientation strategy of blind flies is the same as wild-type flies? It is known that vision plays an important role for flying *Drosophila* to pinpoint the location of a food source (Saxena and Sane, 2018). Blinding walking flies might therefore affect their natural responses to olfactory simulations.

Reviewer #3:

1) The modeling part should be strengthened: while it is understood that the model is empirical, some more intuition and details would definitely help.

Two different filters are being used for the ON and the OFF. What is the logic and the reason for these two different choices, namely the subtraction in the OFF? It would also be useful if the filters in Figure 4 looked more what they really are, to avoid possible confusion.

It is mentioned that the distribution of angular velocities is not matched to the data. How is this discrepancy rationalized? There are also other discrepancies, e.g. in the timing in Figure 4. While the desire to highlight positive results is comprehensible, those limitations are equally important for a global understanding, and should be mentioned in the discussion.

The model has a number of parameters that are empirically tuned. Any insight and comment on how they are affected if the source intensity and/or the wind velocity and/or the fluctuation level are modified would be useful, and again help the reader rationalize the empirical findings.

How is the decrease in Figure 3A-B rationalized?

The finding that local search is not influenced by the wind is particularly interesting. What does that imply for the turns that the fly is making? Are they oriented with respect to the exit direction of the insect from the plume?

2) The presentation and the discussion on conditions of the walking flies' olfactory searches need improvement and clarifications.

Odor detections and turbulence at 50m (in the atmospheric boundary layer) are not like at 50cm from a source (in a boundary layer). It should be stressed that searches for walking flies are happening at sub-meter distances and clear distinctions with the situation of moths (where searches are over tens or even hundreds of meters) should be made. This is not done in the present version, where the introduction focuses on laminar vs turbulent conditions only, and there is no mention of this basic fundamental difference between moths and walking flies.

The last paragraph of subsection “Temporal features of odor driving ON and OFF behaviors”, is another illustration of the above point. The discussion proceeds on the basis that odor detections experienced by moths and walking flies are similar, which is far from being the case. At tens/hundreds of meters, the size of the odor filaments is very small, and the plume is broken up, contrary to what happens at 50cm. A sense of the differences between typical durations of whiffs for the two situations is provided by the comparison between Figure 3 and Figure 4 in Phys. Rev. X, 041015 (2014). The peaks in the two distributions differ by two orders of magnitude, which is theoretically understood by the very different properties of transport at those distances.

Another difference is that at 50cm, inside the plume the level of concentration is fluctuating but detections are essentially continuous. The only way to lose the signal is to cross the well-defined border of the plume (which is again not the case at tens of meters, where the plume is genuinely broken up). This makes that the main issue of the search, even in the presence of turbulence, is to keep contact with the continuous trail, i.e. not that different from the tracking of a laminar tube. While the continuity of detections is somehow witnessed by Figure 6J, it should be made more explicit. It would also be useful to add to Figure 6A a plot of the signal on the scale of the detection level K_d_ defined in Eq. (7) (which is what really matters). Finally, the work Curr. Biol. 26, 1261, 2016 should be mentioned as it also deals with similar distances (for rodents) and shows that there is enough signal to permit even gradient-climbing searches.

[Editors' note: further revisions were requested prior to acceptance, as described below.]

Thank you for resubmitting your work entitled "Elementary sensory-motor transformations underlying olfactory navigation in walking fruit flies" for further consideration at *eLife*. Your revised article has been favorably evaluated by Eve Marder (Senior Editor), a Reviewing Editor, and two reviewers.

The manuscript has been improved but there are some remaining issues that need to be addressed before acceptance, as outlined below:

Please respond to the reviewer comments below. These should be quick to accomplish and will not require re-review.

Reviewer #2:

The authors have revised their manuscript in a way that addresses the concerns I (reviewer 1) raised in my first report. The technical limitations that prevented them to carry out unilateral olfactory stimulation experiments are reasonable. Instead, the authors take advantage of their computational simulations to determine the potential contribution of bilateral sampling to the navigational performances. The results described in the new Figure 7 represents a great addition to the manuscript – I praise the authors for including this new material. What I am not entirely following is the choice of *w1118* as the background of the sighted control. *w1118* is obviously the right genetic background for the "sighted control", but the *w1118* allele affects the visual system of the fly. Since the white-eyed phenotype was not used to keep track of the *norpA* mutation, I am unsure why *w1118* was used as the background of the tested flies in the first place. This choice cannot (and might not have to) be changed, but the authors should consider justifying the use of *w1118* background in the Materials and methods section (other readers might be puzzled as well). While it is true that the sighted control demonstrates ON and OFF responses, blindness produces significant behavioral differences that the reader should keep in mind. It might be worth mentioning this point again in the Discussion section.

Reviewer #3:

The authors have taken into account most previous comments in a satisfactory way. The logic of the choice of the parameters, filters, etc., is more transparent. The Discussion section has improved and the Introduction makes clearer the conditions of the search.

The authors have not modified Figure 6. I disagree with their argument since the mean profile is quite relevant at those distances, as their work also shows. However, this is not my paper and since the point can be grasped alternatively (even though less transparently), I shall not insist.

The authors have included the comment: "Second, plumes developing near a boundary are broad and relatively continuous, while those in open air, particularly at the long distances covered by moths, are much more intermittent", which is good. Adding also a reference to older experimental data, e.g. Yee et al.,(1993), would be useful.

---

## [Author Response]

Summary:In this manuscript, the authors present a creative analysis of fly walking navigation in response to an attractive odor (apple cider vinegar, ACV) in a mini wind tunnel that creates a boundary layer odor plume in which the fly is constrained to walk. Flies were placed in the rectangular arenas, where they were exposed to a constant flow of filtered, humidified air, defining the wind direction. Into this airflow pulses of odor were injected with rapid onset and offset kinetics, producing a front of odor that was transported down the arena at 11.9 cm/s. The odor dynamics were monitored by a photo-ionization detector (PID). They also constructed and tested flies in larger wind tunnels capable of delivering a turbulent boundary layer odor plume after the mini wind tunnel results. They also developed a simple computational model of how odor dynamics and wind direction influence changes in forward and angular velocity to simulate results in both types of wind tunnels. All flies tested were genetically blinded. Behavioral results with odor pulses show that flies have two basic responses with odor onset flies orient upwind and increase their ground speed (on response), and with odor offset, they reduce their ground speed and increase their rate of turning (off response). By mechanically blocking antennal wind sensation, they show that antennal mechanosensation is required for the directional components of these behaviors – on response, while odor offset is sufficient to induce changes in ground speed and turning – off response. Using time-varying odor stimuli, they show that the responses, on and off, low pass filter the odor dynamics and fitting their model they estimate 0.72 s and 4.84 s time constants respectively and that both responses show adaptation to odor with a time constant around 10 s. With the model they show that simulated flies respond similarly to the mean of all flies and that by simply changing the gains of the on and off responses they can capture individual variation about this mean. They then test files in the turbulent plume of the larger wind tunnel and find that their model can likewise capture their behavior including failures to find the odor source. They conclude that integration over time may be a useful computational strategy for navigating in a boundary layer plume, allowing flies to head upwind more continuously in the face of odor fluctuations, and to generate re-orientations clustered at the plume edges. The impact of the paper is in reducing walking navigation in a turbulent boundary layer to two basic stimulus driven responses one bi-modal (wind and odor on response) and one unimodal (odor off response) and providing a simple model to show how these responses interact in real odor plumes. They recognize that their analyses/conclusions are a first order approximation and that other variables/responses can be considered in future but nevertheless, the work is a major step forward in walking odor navigation and should be of wide interest in the behavioral neuroscience community.

We thank the reviewers for their kind words about our study. Our intent in these miniature wind tunnels was to create an odor “front” in which the time of odor onset and offset could be precisely known, and flies within the odor would be unable to move themselves out of it, not to create a boundary layer odor plume as previous studies have done. We have added text to the Results section to try to make the logic of the arena design clearer:

“Flies were placed in rectangular arenas, where they were exposed to a constant flow of filtered, humidified air, defining the wind direction. Into this airflow we injected pulses of odor with rapid onset and offset kinetics, producing a front of odor that was transported down the arena at 11.9 cm/s”.

Essential revisions:1) As stated in the shortened review of reviewer 2, the authors should consider experiments in which one antenna is blocked; mechanoreceptors and potentially also olfactory receptors. These experiments will address the contribution of bilateral antennal sampling, which is known to be important in flies and other insects.

We now address the contribution of bilateral sampling by adding a bilateral term to our computational model. These results are quite interesting (see below); and we believe strengthen our manuscript. We have not included an experimental analysis of bilateral sampling because our apparatus, which attempts to create an abrupt, spatially uniform step in odor concentration, would likely minimize the impact of bilateral sampling on navigation. Instead we draw on the results of previous studies to constrain our model.

The new results are presented in Figure 7, now added to the main Results section. Briefly, we find that if the gain on bilateral sampling is sufficiently high, then adding bilateral sampling significantly improves the performance of the model. This improvement depends on the correct orientation of the sensors; if the sensors are swapped, then the model performs more poorly than without bilateral sampling. Finally, we show that bilateral sensing alone is not sufficient to find the source: in the absence of wind sensing, almost no model flies find the source. These results suggest that wind sensing and bilateral sensing can work cooperatively to promote source localization. This is surprising, given that (1) there is no explicit interaction between the contributions of each form of sensing in our model (they sum linearly), and (2) previous studies have suggested that there is insufficient spatial information for bilateral sensing to contribute to source finding. We discuss these results in the light of past literature in the revised Discussion section.

An issue in this analysis is estimating the gain of bilateral sensing. Based on the studies of Borst and Heisenberg, (1982), and Gaudry and Wilson, (2013), in which different odor concentrations were presented to a tethered fly on a ball, we have estimated that the gain for bilateral sensing in walking flies is around ~40°/sec for a maximal concentration difference between the antennae. In both of these studies, rather constant concentration differences were presented. In our model, we explored a wide range of gain values around this (from 0-300°/sec) that are consistent with the range of plausible instantaneous angular velocities for a walking fly. We found that the contribution of bilateral sensing depended strongly on this gain term. Our modeling results suggest that it would be interesting to measure the dynamic gain of bilateral sensing directly. However, we feel that this lies beyond the scope of the present work, as it would require an entirely different experimental strategy: presenting a tethered walking fly with fluctuating stimuli in which small concentration differences occur across the antennae. These points are added to the revised Discussion section (new final paragraph).

In addition, we have expanded our data set of flies in the New York wind-tunnel (presented in Figure 6) from 14 flies to 66. We have updated Figure 6I and its legend accordingly, as well as the text in the corresponding parts of the Results (subsection “Behavior of real and model flies in a turbulent environment”). These changes appear highlighted in blue like all other changes in the edited manuscript.

2) There are concerns that using only blind flies may influence the navigation strategy observed. Any data from sighted flies that could be included should be, or the potential implications of blindness should be more extensively discussed.

We have now added data from sighted flies (Figure 1—figure supplement 3) in the same genetic background (*w1118* 5905). These data show that sighted flies also exhibit ON and OFF responses to odor, although the offset response (increase in angular velocity) is smaller compared to responses measured in genetically blind (*norpA*) flies.

3) As stated in the shortened review of reviewer 3, the model should be better clarified and the parameters better rationalized.

We have revised the Results section and Materials and methods section to clarify the nature of the model parameters and to explain the procedure by which they were tuned.

4) The Discussion section should more extensively discuss the type of navigation the flies are doing here better in the context of flight navigation.

We have revised the Discussion section to clarify the differences between plumes generated near a solid boundary (the typical environment of a walking fly) and those away from a surface (the typical environment of a flying insect).

Reviewer #2:- In past work, several labs have reported the contribution of bilateral sampling (stereo-olfaction), from the pioneering work of Axel Borst in 1983 to the recent optogenetic-stimulation experiments conducted by Gaudry and Wilson. The fact that this aspect of the navigation mechanism is not addressed in the present manuscript represents a weakness. It seems that the author could have tested relatively easily the behavior of unilaterally wind-blind flies. They could then ask whether up-wind surges are possible with a single functional antenna, whether unilaterally wind-blind flies display a turn bias, etc. Likewise, unilateral olfactory impairment could have been produced through mechanical impairments (Duistermars and Frye, 2009). Combining these results in the model would nicely complete the present analysis. Although this weakness is minor, the authors are encouraged to address it.

We thank the reviewer for raising the issue of bilateral sampling. As mentioned above under “Essential Revisions,” we have now added a bilateral olfactory sampling term to our model and report the results of these simulations in a new Figure 7. As described there, we find that the role of bilateral sensing depends on the gain (how much does that animal turn given a certain difference in concentration). Based on the Borst, 1982 and Gaudry, 2013 studies we estimated that flies turn at a rate of ~40°/sec when a maximal concentration difference is applied across the antennae. In this case we found that bilateral sensing contributed little to the probability of finding the odor source. However, if we increased the gain to 300°/sec, then bilateral sensing significantly improved performance. We show that this improvement requires the correct orientation of the sensors, and also that bilateral sampling is insufficient to find the source in the absence of wind sensing. We have added material to the Discussion section describing the relationship of these findings to pervious work and their implications for future studies. We believe these computational results significantly improve the manuscript and thank the reviewer for encouraging us to pursue them.

Additionally, we have performed unilateral wind-blinding experiments and found that they result in a reduced upwind velocity. However, we believe these experiments speak most directly to the question of how and where in the brain wind direction is computed from the movements of the two antennae, which is the subject of a manuscript we are currently preparing, rather than the question of what algorithms flies use to navigate towards an odor source, given that they can use their antennae to sense wind direction. We would therefore prefer to reserve these data for the other manuscript.

We also made an attempt to perform unilateral olfactory impairment, however, we concluded that these experiments would not provide much insight. First, we found that it was not possible to completely cover all olfactory sensilla and still leave antennal movement intact, permitting full wind-sensing. Second, our arenas were designed to minimize spatial differences in odor concentration; thus there are no systematic spatial concentration differences to be detected across the antennae. In wind-blind flies we detected no turning towards the side the was hit first by the odor front, suggesting that flies are unable to use these brief timing differences for navigation. Finally, in our arenas flies can walk either on the floor or on the ceiling. This makes it impossible for us to detect a turning bias towards the intact antenna, because we cannot distinguish whether the fly is right side up or upside down.

- The use of blind (norpA) mutants in the olfactory navigation experiments is sensible. I was nonetheless wondering whether the authors have any evidence that the orientation strategy of blind flies is the same as wild-type flies? It is known that vision plays an important role for flying Drosophila to pinpoint the location of a food source (Saxena and Sane, 2018). Blinding walking flies might therefore affect their natural responses to olfactory simulations.

We have included a new supplementary figure (Figure 1—figure supplement 3) with behavioral data from sighted flies of identical genetic background, and added text describing these data to subsection “ON and OFF responses to odor in a miniature wind-tunnel paradigm”:

“Sighted flies of the same genetic background also showed ON and OFF responses (Figure 1—figure supplement 3), with increases in upwind velocity and groundspeed during odor, and increases in angular velocity and decreased groundspeed after odor offset. However, the increase in angular velocity appeared to be weaker, on average, in sighted flies.”

Reviewer #3:1) The modeling part should be strengthened: while it is understood that the model is empirical, some more intuition and details would definitely help.Two different filters are being used for the ON and the OFF. What is the logic and the reason for these two different choices, namely the subtraction in the OFF? It would also be useful if the filters in Figure 4 looked more what they really are, to avoid possible confusion.

We have updated the text and revised Figure 4 to clarify the logic behind these choices (subsection “Phenomenological models of ON and OFF responses”). Our reasoning was as follows: We chose two different filter shapes to model these responses as they occur at different phases relative to the odor stimulus: the ON response occurs during the odor and looks like a smoothed version of the odor stimulus, while the OFF response occurs after the odor, and reflects a decrease in odor concentration. Therefore, we modeled the ON response as an integrating (low-pass) filter and the OFF response as a differentiating (high-pass) filter. In principle, many filter shapes could give rise to these two shapes. For simplicity, we chose to parameterize the integrating filter by a single time constant which describes the time interval over which the local odor concentration is smoothed to generate the response. The differentiating filter computes the difference between the current odor concentration and the past odor concentration and is parameterized by two-time constants, one which specifies the interval over which the current concentration is computed, and one of which specifies the interval over which the past concentration is computed. We have updated the text of the results to try to make the logic behind these choices clearer (subsection “Phenomenological models of ON and OFF responses”). We have also updated the filters in Figure 4 to more closely resemble those used. The new text is:

“We then tested four models for the ON response: one with adaptive compression followed by a low-pass filter (``ACF''), one with filtering followed by adaptive compression (``FAC''), and the same models without adaptation (``CF'' and ``FC'' respectively). We note that the FC model, with filtering followed by a fixed nonlinearity, is most similar to traditional linear-nonlinear models. For simplicity, we parameterized the low-pass filter by a single time constant tauON, that describes the amount of smoothing seen in the response (Methods).”

“In this case, the adaptive compression step was the same, but we used a differentiating filter instead of a low-pass filter, to generate responses when the odor concentration decreases from a previously high level. This filter was parameterized by two time constants, tauOFF1 and tauOFF2, that describe the time intervals over which the current and past odor concentrations are measured (Figure 4F, Methods).”

It is mentioned that the distribution of angular velocities is not matched to the data. How is this discrepancy rationalized? There are also other discrepancies, e.g. in the timing in Figure 4. While the desire to highlight positive results is comprehensible, those limitations are equally important for a global understanding, and should be mentioned in the discussion.

We have added a mention of the discrepancy between the model predictions and the timing of response in Figure 4 to subsection “Modeling olfactory search behaviour”.

“There are also discrepancies between our model predictions and the timing of responses near odor onset (particularly in the frequency sweep responses) that might reflect the simplicity of the filter model used or might reflect real variability in the latency of flies to respond to odor.”

The mismatch between the real angular velocity distribution and the one we used in the model is interesting. We have spent a great deal of time trying to understand this but have not been successful so far. As noted in the Materials and methods section when we tried to draw angular velocities from the real distribution, we generated trajectories that were much more jagged than what we observed experimentally. This implies that one of the assumptions of our model, for example the assumption that turns are drawn randomly, or that angular velocity is chosen independent of forward velocity, must be incorrect. We have added a note to this effect to the Materials and methods section. A recent study (Kim and Dickinson, 2017) reached similar conclusions when analyzing turns generated upon removal of flies from food. That study concluded that turns must be driven by path integration but did not provide a generative model for producing realistic turns. Given this state of the field, we thought it best to be upfront about this limitation of the model, and to leave the question of how to model turning more realistically as a problem for the future.

The model has a number of parameters that are empirically tuned. Any insight and comment on how they are affected if the source intensity and/or the wind velocity and/or the fluctuation level are modified would be useful, and again help the reader rationalize the empirical findings.

We have added material to the Materials and methods section and Results section to provide further information on parameter tuning. We have revised our description of the navigation model parameters in the Results section to explain that these controls how strongly a change in odor concentration affects behavior:

“This navigation model is parameterized by six coefficients (K1-K6) that determine the strength with which the ON and OFF functions modulate ground speed, turn probability, and the drive to turn up- or downwind. For example, K1 determines how much the forward velocity increases when the ON function increases by a specific amount.”

We have also added details to the Materials and methods section on how these parameters were tuned:

“To estimate values for the coefficients *κ*1 to *κ*6 we ran simulations of the model using a 10 s odor pulse and adjusted parameters sequentially so that analysis of the model outputs matched as closely as possible the response of real flies shown in Figure 5E. We first adjusted *κ*1 and *κ*2 to match the ground speed. Then we adjusted *κ*5 and *κ*6 and matched the upwind velocity. Finally, we adjusted *κ*3 and *κ*4 to match turn probability. We matched the theoretical turn probability on the model (the rate governing the Poisson variable rho(t) rather than the turn probability extracted from the model trajectories”

A single set of parameters was obtained by tuning to match the “mean fly” from Figure 1. The sensitivity of the behavior to odor concentration and fluctuation level were explored in Figure 3. These experiments were designed to ask how behavioral responses change in the face of different odor concentrations and fluctuation levels, including both un-natural fluctuations (square pulse, frequency sweep) and natural ones (plume walk). To the extent that the ON and OFF models capture these dynamics (Figure 4), we believe that we can conclude that responses to all of these stimuli are equally well captured by the same model.

We have not included data on altered wind velocity here but have run some preliminary experiments of this type. In these experiments we found little effect of halving the wind velocity, once we accounted for the fact that odor was transported down the arena at half the speed. ON and OFF responses are present and have very similar magnitude, with a slightly smaller modulation of speed and slower dynamics of the onset responses, perhaps due to a higher difficulty to estimate wind direction. It would be interesting in a future study to explore the effects of wind velocity more systematically.

How is the decrease in Figure 3A-B rationalized?

This is a curious, but we think robust, finding. One possible explanation is that at higher intensities, additional receptors/glomeruli are recruited that have an inhibitory effect on the behavioral responses described here. Inhibitory effects of low-affinity glomeruli on odor attraction have been described in Semmelhack and Wang, 2009. We have added this citation as follows:

“A decrease in response at the highest intensities could arise from inhibitory glomeruli that are recruited at higher odor intensity, as has been described in (Semmelhack, 2009)”.

The finding that local search is not influenced by the wind is particularly interesting. What does that imply for the turns that the fly is making? Are they oriented with respect to the exit direction of the insect from the plume?

We agree that this aspect of the local search is interesting and makes it distinct from casting behavior described in flying flies, in which a clear orientation perpendicular to the wind has been observed. We include some additional analysis of turns as a function of orientation within the plume here; however, as these results do not substantially alter the conclusions of the manuscript, we have not included them in the text.

To look at the relationship between exit direction and turning more closely, we calculated the average angular velocity as a function of orientation with respect to the wind, similar to what we did in the “D-functions” in Figure 5D. We performed this analysis in two ways, first calculating angular velocity over some interval as a function of heading at the time of odor offset (Author response image 1, left), and also calculating angular velocity as a function of instantaneous heading over the same intervals (Author response image 1, right). Each upper graph shows (mean ± SEM) for control flies for three different time windows immediately after odor offset (400, 1000 and 2000 ms).

In the upper-left hand plot we see a very slight “upwind” shape that is strongest for the shortest interval and becomes weaker. We think this likely reflects the fact that flies continue to orient upwind for a short period after odor offset. In the upper-right hand plot we see the gradual emergence of a “downwind” shape only between 1-2 sec. The bottom plot shows the polar distributions of flies’ movements during the same windows described in the upper plots, plus the distributions during the whole ON period and the first 10 seconds of the OFF response. It shows more clearly how the average movement immediately after odor offset transitions from the ON to the OFF behavior. We interpret these analyses to mean that flies continue to orient upwind briefly after odor offset, and then gradually switch to downwind orientation, as predicted by our model.

2) The presentation and the discussion on conditions of the walking flies' olfactory searches need improvement and clarifications.Odor detections and turbulence at 50m (in the atmospheric boundary layer) are not like at 50cm from a source (in a boundary layer). It should be stressed that searches for walking flies are happening at sub-meter distances and clear distinctions with the situation of moths (where searches are over tens or even hundreds of meters) should be made. This is not done in the present version, where the introduction focuses on laminar vs turbulent conditions only, and there is no mention of this basic fundamental difference between moths and walking flies.

We did draw this distinction in the Introduction, where we stated that a plume developing near a solid boundary (the environment in which a walking fly will search) is distinct from an airborne plume. We have revised this sentence (to further differentiate the environment of walking versus flying searchers:

“A walking fly in nature will encounter an odor plume that is developing close to a solid boundary. Such plumes are broader, exhibit slower fluctuations, and allow odor to persist further downwind from the source, compared to the airborne plumes encountered by flying organisms (Crimaldi, 2001; Crimaldi, 2002; Webster, 2001). Navigational strategies in these two environments might therefore be different.”

The last paragraph of subsection “Temporal features of odor driving ON and OFF behaviors”, is another illustration of the above point. The discussion proceeds on the basis that odor detections experienced by moths and walking flies are similar, which is far from being the case. At tens/hundreds of meters, the size of the odor filaments is very small, and the plume is broken up, contrary to what happens at 50cm. A sense of the differences between typical durations of whiffs for the two situations is provided by the comparison between Figure 3 and Figure 4 in Phys. Rev. X, 041015 (2014). The peaks in the two distributions differ by two orders of magnitude, which is theoretically understood by the very different properties of transport at those distances.

We have revised this paragraph to emphasize the longer distances involved in moth searching:

“Why might olfactory behavior in walking flies reflect integration of olfactory information over time while upwind flight in moths appears to require a rapidly fluctuating stimulus? Several possibilities are worth considering. One is that the temporal demands of walking differ from those of flight. A flying moth travels at much faster speeds and over longer distances than a walking fly and will therefore traverse a plume in less time. Second, plumes developing near a boundary are broad and relatively continuous, while those in open air, particularly at the long distances covered by moths, are much more intermittent (Crimaldi, 2001; Celani, 2014), again making detection of the plume edge potentially more important than responding rapidly to each plume encounter.”

Another difference is that at 50cm, inside the plume the level of concentration is fluctuating but detections are essentially continuous. The only way to lose the signal is to cross the well-defined border of the plume (which is again not the case at tens of meters, where the plume is genuinely broken up). This makes that the main issue of the search, even in the presence of turbulence, is to keep contact with the continuous trail, i.e. not that different from the tracking of a laminar tube. While the continuity of detections is somehow witnessed by Figure 6J, it should be made more explicit. It would also be useful to add to Figure 6A a plot of the signal on the scale of the detection level K_d_ defined in Eq. (7) (which is what really matters). Finally, the work Curr. Biol. 26, 1261, 2016 should be mentioned as it also deals with similar distances (for rodents) and shows that there is enough signal to permit even gradient-climbing searches.

We have created the plot suggested, showing the time-averaged odor plume after being processed by the compressive nonlinearity using a K_d_ of 0.01, and present it here (Author Response Image 2). However, we think it may be misleading with regard to understanding plume structure and navigation algorithms, as it relates to the time-averaged plume, and not the dynamics the fly actually encounters. We therefore do not think it should be included in the manuscript. We note that the effects of compression are already visible in Figure 6G in the context of the dynamic signal encountered by the navigating model fly.

In the revised paragraph above we have noted that the boundary layer plume is “relatively continuous.” We also note in the introduction that boundary layer plumes “allow odor to persist further downwind from the source” — a reference to the decreased intermittency and greater continuity seen in these plumes.

We have added a citation to Gire, 2016 to the introduction where we mention the differences between plumes near a solid boundary versus in open air and also to the Discussion section where we note that the boundary layer plume we measured appears to contain useful spatial information for navigation.

**Author response image 2. respfig2:** 

[Editors' note: further revisions were requested prior to acceptance, as described below.]

The manuscript has been improved but there are some remaining issues that need to be addressed before acceptance, as outlined below:Please respond to the reviewer comments below. These should be quick to accomplish and will not require re-review.Reviewer #2:The authors have revised their manuscript in a way that addresses the concerns I (reviewer 1) raised in my first report. The technical limitations that prevented them to carry out unilateral olfactory stimulation experiments are reasonable. Instead, the authors take advantage of their computational simulations to determine the potential contribution of bilateral sampling to the navigational performances. The results described in the new Figure 7 represents a great addition to the manuscript – I praise the authors for including this new material. What I am not entirely following is the choice of w1118 as the background of the sighted control. w1118 is obviously the right genetic background for the "sighted control", but the w1118 allele affects the visual system of the fly. Since the white-eyed phenotype was not used to keep track of the norpA mutation, I am unsure why w1118 was used as the background of the tested flies in the first place. This choice cannot (and might not have to) be changed, but the authors should consider justifying the use of w1118 background in the method section (other readers might be puzzled as well). While it is true that the sighted control demonstrates ON and OFF responses, blindness produces significant behavioral differences that the reader should keep in mind. It might be worth mentioning this point again in the discussion.

Since we had run most of our behavior in *norpA w1118* flies, we thought this was the best control for the effects of *norpA*. The *w1118* 5905 strain was chosen because it exhibits robust walking behavior and has previously been used for a number of behavioral studies, especially examining sleep. In addition, most transgenic strains as built in a *w1118* background.

We have added the following text to the Materials and methods section:

We used only *w1118 norpA^36^* flies for behavior. For this reason, we used *w1118* flies as ‘sighted’ controls, although the *w1118* allele does affect vision as well.

We have added the following text to the Discussion section:

OFF responses were weaker in flies lacking the *norpA^36^* allele, suggesting that vision may be able to substitute to some degree for search behavior, or that the *norpA^36^* allele itself promotes more vigorous searching.

Reviewer #3:The authors have taken into account most previous comments in a satisfactory way. The logic of the choice of the parameters, filters, etc., is more transparent. The Discussion section has improved and the Introduction makes clearer the conditions of the search.The authors have not modified Figure 6. I disagree with their argument since the mean profile is quite relevant at those distances, as their work also shows. However, this is not my paper and since the point can be grasped alternatively (even though less transparently), I shall not insist.

We thank the reviewer for respecting our point of view. We still think the mean profile is somewhat misleading, as the fly never encounters this directly.

The authors have included the comment: "Second, plumes developing near a boundary are broad and relatively continuous, while those in open air, particularly at the long distances covered by moths, are much more intermittent [20] [17].…", which is good. Adding also a reference to older experimental data, e.g. Yee et al., (1993), would be useful.

We have added the suggested citation.